# Microtubule association induces a Mg-free apo-like ADP pre-release conformation in kinesin-1 that is unaffected by its autoinhibitory tail

J. Atherton [1,6] ✉, M. S. Chegkazi[1,5,6], M. Leusciatti[2,3], M. Di Palma [2],
E. Peirano [2], L. S. Pozzer [2], M. V. A. Meli[3], S. Pasqualato [4], T. Foran[1],
G. Morra [3] & R. A. Steiner [1,2] ✉

Kinesin-1 is a processive dimeric ATP-driven motor that transports vital intracellular cargos along microtubules (MTs). If not engaged in active transport, kinesin-1 limits futile ATP hydrolysis by adopting a compact autoinhibited conformation that involves an interaction between its C-terminal tail and the N-terminal motor domains. Here, using a chimeric kinesin-1 that fuses the N-terminal motor region to the tail and a tail variant unable to interact with the motors, we employ cryo-EM to investigate elements of the MT-associated mechanochemical cycle. We describe a missing structure for the proposed two-step allosteric mechanism of ADP release, the ATPase rate limiting step. It shows that MT association remodels the hydrogen bond network at the nucleotide binding site triggering removal of the $Mg^{2+}$ ion from the $Mg^{2+}$-ADP complex. This results in a strong MT-binding apo-like state before ADP dissociation, which molecular dynamics simulations indicate is mediated by loop 9 dynamics. We further demonstrate that tail association does not directly affect this mechanism, nor the adoption of the ATP hydrolysis-competent conformation, nor neck linker docking/undocking, even when zippering the two motor domains. We propose a revised mechanism for tail-dependent kinesin-1 autoinhibition and suggest a possible explanation for its characteristic pausing behavior on MTs.

Kinesin-1 (also known as conventional kinesin) is an essential ATP-driven molecular motor of the kinesin superfamily capable of processive movement as stepping dimers toward the (+)-end of polarized microtubules (MTs)[1]. This enables the transport of a range of cargoes required for cellular function, including mitochondria[2,3], synaptic vesicles and plasma membrane precursors[4], lysosomes[5,6], and mRNAs[7].

Several viruses also hijack the MT transport machinery of the host cell to facilitate their replication and spread[8].

The two motor-bearing heavy chains (Kif5) are typically associated with two light chain (KLC) adapters that form the tetrameric kinesin-1 complex. KLCs often mediate the interaction with the proteinaceous components of cellular cargoes by recognizing short-linear

[1]Randall Centre for Cell and Molecular Biophysics, King's College London - New Hunt's House, Guy's Campus, London, UK. [2]Department of Biomedical Sciences, University of Padova, Padova, Italy. [3]Istituto di Scienze e Tecnologie Chimiche 'G. Natta' SCITEC, Consiglio Nazionale delle Ricerche, Milano, Italy. [4]Human Technopole, Milano, Italy. [5]Present address: ELIXIR Hub, South Building, Wellcome Genome Campus, Hinxton, Cambridge, UK. [6]These authors contributed equally: J. Atherton, M. S. Chegkazi. ✉e-mail: joseph.atherton@kcl.ac.uk; roberto.steiner@kcl.ac.uk

motifs (SLiMs)[9–11] and other ordered regions[12]. However, they are not always required, as transport of mitochondria and mRNA granules has been shown to be KLC-independent[13,14]. In the case of mitochondria, KLCs even negatively impact the recruitment of the necessary Milton/Miro adapter complex that binds directly to Kif5[13].

Kif5 contains a N-terminal motor domain (also known as the head) that generates force and processive stepping via Mg²⁺-dependent ATP

hydrolysis and exchange, multiple coiled-coil (CC) regions with breaks (stalk) that mediate dimerization, and an unstructured C-terminus (tail) involved in autoinhibition (Fig. 1a). ATPase activity is strongly activated by MT binding that induces conformational changes in the motor stimulating both ATP hydrolysis and ADP release steps[15–20]. One of the key structural elements undergoing transitions that are coupled to the nucleotide state is the neck linker, that connects the motor

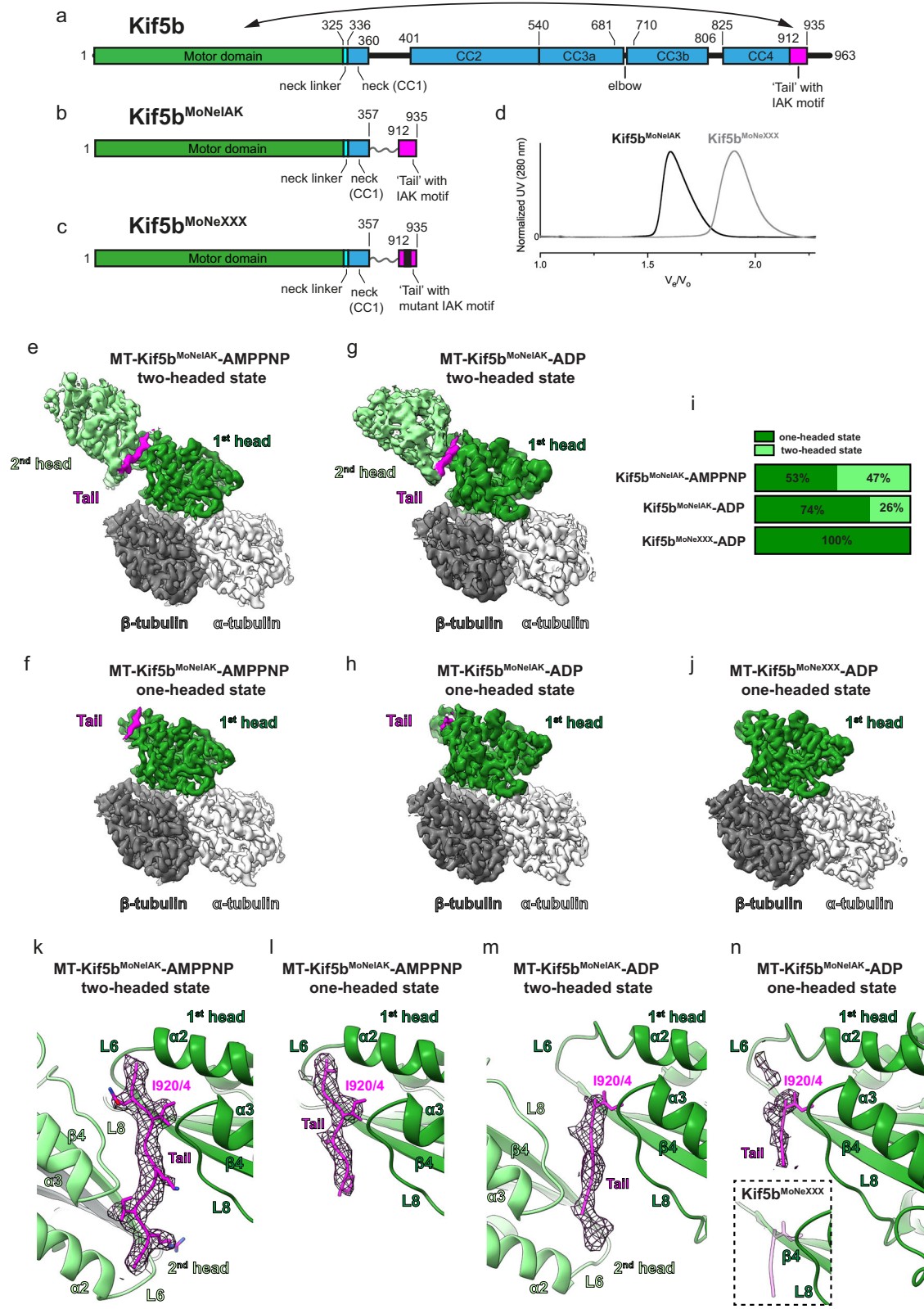

**Fig. 1 | The Kif5 tail IAK motif interacts with MT-bound one-headed and two-headed motors and is required for motor zippering. a–c** Schematics showing the domain organization of **a** full-length Kif5b and of **b** Kif5b[MoNeIAK], and **c** Kif5b[MoNeXXX] chimeras with the mutated IAK motif highlighted in black. Residue numbers are shown for full-length Kif5b, and this numbering is kept for the chimeras. Kif5b[MoNeIAK] includes the N-terminal motor domain (head), neck linker and first coiled-coil (neck and CC1) fused by a flexible linker to amino acids 912–935 of the tail domain, containing the IAK motif. **d** Normalized size-exclusion chromatography (SEC) traces for Kif5b[MoNeIAK] and Kif5b[MoNeXXX] ($V_e$ = elution volume, $V_0$ = exclusion volume). Kif5b[MoNeIAK] and Kif5b[MoNeXXX] elute as dimers and monomers, respectively. Source data are provided as a Source Data file. **e, f** Unsharpened cryo-EM reconstructions of the **e** two-headed and **f** one-headed states of the MT's αβ-tubulin heterodimer-Kif5b[MoNeIAK] asymmetric unit in the presence of AMPPNP. **g, h** Unsharpened cryo-EM reconstructions of the **g** two-headed and **h** one-headed states of the MT's αβ-tubulin heterodimer-Kif5b[MoNeIAK] asymmetric unit in the presence of ADP.

**i** Schematics highlighting the percentage occupancy of one and two-headed states in Kif5b[MoNeIAK] and Kif5b[MoNeXXX] datasets. **j** Overall structure of the MT's αβ-tubulin heterodimer-Kif5b[MoNeIXXX] asymmetric unit, for which only a one-headed state was observed. **k, l** Cryo-EM density for the Kif5b[MoNeIAK] tail (mesh) in the MT-Kif5b[MoNeIAK]-AMPPNP **k** two-headed and **l** one-headed states. **m, n** Cryo-EM density for the Kif5b[MoNeIAK] tail (mesh) in the MT-Kif5b[MoNeIAK]-ADP **m** two-headed and **n** one-headed states. In (**k–n**), the position of key isoleucine I920/4 and side chains for well-resolved tail residues are shown. L6 and L8 indicate loop 6 and loop 8, respectively. The inset in (**n**) shows the MT-Kif5b[MoNeXXX]-ADP head in green, with the potential location of the tail by displaying the MT-Kif5b[MoNeIAK]-ADP tail model in semi-transparent magenta. In this inset, MT-Kif5b[MoNeXXX]-ADP density is shown at the same threshold and with the same filtering as Kif5b[MoNeIAK]-ADP density shown in the main panel (demonstrating no density for the tail). (**e–h**) and (**j**) display unfiltered density. (**k–n**) show LocScale filtered density.

domain to CC1 (Fig. 1a). This element, which is critical for force generation under physiological conditions[21], goes from being docked along the core motor domain in the ATP-bound state to being flexible in both the ADP-bound and nucleotide-free states[15,20,22–28]. For kinesin-1, MT binding speeds up the rate-limiting step of ADP release more than ~104 fold[16]. The structural mechanisms governing MT-stimulated ADP release have been disputed for decades and are still unclear[19,23,26,29,30].

Another key aspect of kinesin-1's mechanochemical cycle is its autoinhibition. To limit wasteful ATP hydrolysis and futile runs, cargoless Kif5 adopts a compacted autoinhibited conformation that involves an interaction between the motor domain and the unstructured C-terminal tail centered on its IAK (isoleucine-alanine-lysine) SLiM[31] (hereafter just IAK motif, Fig. 1a). Binding of a single tail to the kinesin-1 dimer is sufficient for inhibition[32] preventing MT-dependent stimulation of ATPase activity, particularly the acceleration of ADP release[33]. A cryo-EM reconstruction of MT-bound monomeric heads chemically cross-linked to the IAK motif indicated that this inhibitory region engages with the nucleotide-binding switch I motif of loop 9 (see Supplementary Fig. 1a for secondary structure elements of the kinesin-1 motor), with this interaction suggested to directly inhibit ADP release[34]. However, a subsequent crystallographic structure of the autoinhibited ADP-bound *Drosophila* kinesin-1 motor dimer showed the IAK motif zipping the heads by binding away from the nucleotide-binding site[35]. This work proposed a 'double-lockdown' model of autoinhibition, whereby zipping the heads locks them in a conformationally restricted orientation that, in turn, prevents, undocking of the neck linker from the motor that has been suggested to facilitate ADP release[20,24].

Recent studies have shown that critical to the achievement of the compact autoinhibited conformation is a short break (elbow) in CC3 of Kif5 that allows the molecule to fold on itself such that the unstructured C-terminal tail is in proximity of the motors[36–38] (Fig. 1a). These studies suggest that portions of the stalk would reduce MT binding in the folded conformation by sterically interfering with the MT-binding surface of at least one of the motor domains. This is supported by the observation that compared to the motor domain in isolation, MT affinity/landing rate of full-length Kif5 is reduced, with or without KLCs, unless autoinhibition is relieved[33,37,39,40]. However, full-length Kif5 still displays a basal level of landing events and, when MT-associated, reduced motility and longer association/run durations compared to truncated tailless motors[37,40]. The 'stalling' behavior characteristic of full-length kinesin-1 motility has been shown to depend on the tail/IAK motif[37,39,40]. Furthermore, the addition of a tail peptide to dimeric tailless constructs decreases MT-stimulated ATPase and ADP release rates and motility, stalling kinesin in a strongly MT-associated state[41–44]. These observations suggest that at least some aspects of kinesin-1's mechanochemistry are independent of purely steric effects and depend on the direct interaction of the tail with the motor.

Using engineered stalkless chimeric constructs of human Kif5, we present here MT-bound cryo-EM structures that explain key aspects of kinesin-1's mechanochemical cycle and autoinhibition. We have visualized an apo-like transition state of MT-bound kinesin-1 with ADP but lacking $Mg^{2+}$. Our structural analysis shows the MT-induced conformational changes that remove the $Mg^{2+}$ ion, leading, in turn, to ADP destabilization and, ultimately, its release. Moreover, we show that whilst the autoinhibitory tail can crosslink the motors in a zipped dimer, this arrangement is not rigid and, importantly, it does not prevent undocking of the neck linker. Based on our structures, we propose a revised mechanism for kinesin-1 autoinhibition and suggest an explanation for its distinctive stalling behavior.

## Results

### 'Bonsai' kinesin-1 chimeras for cryo-EM studies

It is currently unclear if the association of the autoinhibitory tail affects nucleotide-dependent structural transitions of MT-bound kinesin-1. To explore this using cryo-EM, we designed a stalkless 'bonsai' kinesin-1 chimera, dubbed Kif5b[MoNeIAK] (<u>Mo</u>tor-<u>Ne</u>ck-<u>IAK</u> motif), which fuses the N-terminus of human Kif5b (motor domain up to the end of CC1) to its C-terminal IAK motif via a flexible linker (Fig. 1a, b, and Supplementary Fig. 1a). The linker is sufficiently long to allow the free interaction of the tail with the motor. To specifically address the effects of the IAK motif in an identical background, we also generated an IAK-variant chimera in which we replaced residues 919-QIAKPIR-925 with an unrelated amino acid sequence (TGSTSGT) that was predicted to abrogate motor-tail interactions (Fig. 1c and Supplementary Fig. 1b). We refer to this variant as Kif5b[MoNeXXX]. ATPase measurements of Kif5b[MoNeXXX] compared to isolated motors indicate that the flexible linker does not impact kinetic properties (Supplementary Fig. 2). Size-exclusion chromatography (SEC) runs are consistent with Kif5b[MoNeIAK] and Kif5b[MoNeXXX] dimers and monomers, respectively (Fig. 1d). This supports the notion that the IAK motif is involved in crosslinking the motors in the autoinhibited state[35].

### Tail-bound motors can interact with MTs both as monomers and as zipped dimers

We next sought to assess the structural impact of the tail on Kif5b in ADP and ATP nucleotide states. The ATP analog adenylyl-imidodiphosphate (AMPPNP) or the ADP+Pi analog ADP-AlF$_4$ can be used to stabilize indistinguishable ATP-like conformational states of the Kif5b motor domain on MTs or tubulin[15,23,26]. Therefore, a cryo-EM dataset was collected of taxol-stabilized MTs decorated with Kif5b[MoNeIAK] in the presence of AMPPNP for comparison with the X-ray crystallographic model of ADP-AlF$_4$-bound Kif5b in complex with a tubulin dimer and a DARPin (PDB code 4hna)[15]. Moreover, considering that an atomistic model of the MT-bound Kif5b motor domain in the presence of ADP is not available, we also collected datasets of taxol-stabilized MTs decorated with Kif5b[MoNeIAK] and Kif5b[MoNeXXX] in the

presence of ADP. Data collection statistics for all datasets are reported in (Supplementary Table 1).

MT segments of thirteen protofilaments were processed with a modified version of the MiRP MT processing pipeline[45] (see also the "Methods" section), followed by symmetry expansion, focused extraction, focused 3D classification, and focused 3D refinement on the asymmetric unit. This allowed conformational sorting of individual motor-(αβ)tubulin complexes (Fig. 1e–i, Supplementary Fig. 3). For Kif5b[MoNeIAK], we recognized two distinct populations of MT-bound motors. Irrespective of the nucleotide state, they were present either in a two-headed state with motors zipped by the autoinhibitory tail (Fig. 1e, g) or in a one-headed state with some tail density still visible associated with the MT-bound motor (Fig. 1f, h). The fraction in the two-headed state was more abundant with AMPPNP (47% of the total) compared to ADP (26%) (Fig. 1i). For ADP-Kif5b[MoNeXXX], we only observed one-headed motors and no density for the tail (Fig. 1i, j). Whilst one-headed Kif5b[MoNeXXX] is fully consistent with the monomeric population seen in SEC (Fig. 1d), in the case of Kif5b[MoNeIAK] its one-headed fraction can be rationalized either because of monomerization due to dilution required for cryo-EM grid preparation or because of flexibility in regions of the neck linker and therefore heterogeneity in the position of the CC and distal head (2nd head) compared to the MT-bound one (1st head) making the later invisible due to the averaging process in cryo-EM reconstruction. In all reconstructions, quality and resolution varied within the asymmetric unit (Supplementary Figs. 4–7), with the MT-motor domain interface being resolved best at ~2.8–3 Å resolution and decreasing to around ~3.4–4 Å at the motor tip and tail density. In the two-headed states, the resolution dropped rapidly from ~3.5 Å to ~8 Å for the distal motor head moving away from the dimer interface/tail. This decrease in resolution is consistent with some variability in position or conformation for the distal head.

Overall, the arrangement of the MT-bound two-headed complex observed here is consistent with the X-ray crystallographic structure of the autoinhibited MT-free *Drosophila melanogaster* kinesin-1 motors (PDB code 2y65)[35]. However, structural superposition of our cryo-EM structures with the crystallographic one using one of the motors as a frame of reference (the MT-bound one for the cryo-EM structures) reveals variability in the relative positioning of the distal head. This is quantified by ~14° and ~21° rotations, in opposite directions and around independent axes, for the AMPPNP and ADP and states, respectively (Supplementary Fig. 8). This suggests that, differently from the symmetric crystallographic complex, some variability in the relative orientation of the zippered heads is compatible with the presence of the tail.

In Kif5b[MoNeIAK], density for the tail is visible in all states − though with varying levels of order − in a cleft between loop 8 and a region leading from the C-terminal tip of helix α2 though loop 6 to the N-terminal end of the β4 strand of the core motor domain β-sheet (Fig. 1k–n). This location matches that seen in the MT-free crystallographic structure of autoinhibited *Drosophila* kinesin-1[35]. Although our construct contains a 24 amino acid-long tail region (Fig. 1b), a maximum of nine amino acids (AQIAKPIRP) could be modeled in the two-headed AMPPNP-Kif5b[MoNeIAK] structure (Fig. 1k). Like in the X-ray structure, we could not define a clear directionality for the pseudo-palindromic IAK motif; thus, the peptides were modeled in both polarities (Supplementary Fig. 9a − for clarity, only one peptide is shown in Fig. 1k–n). Sitting at the center of the pseudo-palindrome, K922 has the same location in the complex regardless of polarity with either I920 or I924 embedded in a hydrophobic pocket lined by I130 and F128 of strand β4, I119, Y120 and F116 of helix α2 and C174 of loop 8 of the motor domain (Supplementary Fig. 9a, b). Tail binding is not restricted to the two-headed states that offer pseudo-symmetric stabilization (Fig. 1k, m), as density is clearly present also in the one-headed states (Fig. 1n, l). Whilst for ADP-Kif5b[MoNeIAK] only the IAK tripeptide (KPI in the opposite direction) could be modeled (Fig. 1n), for

AMPPNP-Kif5b[MoNeIAK] the ordered region is longer (AQIAKP or AKPIRP) (Fig. 1l & Supplementary Fig. 9a). Overall, these observations suggest a mechanism for the formation of the autoinhibited zipped dimer in which a 'IAK half-site' binds first to one motor head, leading then to the recruitment of the other motor by avidity.

## Association with MTs removes the $Mg^{2+}$ ion from ADP-bound kinesin-1, resulting in an apo-like intermediate state

Currently, there is no atomistic model of the MT-bound kinesin-1 motor in the ADP state. Thus, our ~2.9 Å resolution ADP-Kif5b[MoNeXXX] structure fills an important gap in knowledge. For this chimera, which lacks the endogenous IAK motif, only one-headed MT-bound motors were observed that are devoid, as designed, of the extra density for the tail at its binding location (Figs. 1j, n, and 2a). Density for the neck linker is also absent (Fig. 2b), indicating that it is undocked and flexible, in keeping with low-resolution reconstructions[20,46]. Noticeably, whilst ADP is present in its binding pocket, density for $Mg^{2+}$ is absent (Fig. 2c), despite its clear presence at the N-site of α-tubulin (Supplementary Fig. 10a). Compared to the crystallographic ADP-bound Kif5b motor structure solved in the absence of MTs (PDB code 1bg2)[47] (Fig. 2d), here, loop 9 that harbors the 'switch I' motif, unfurls from a short helical segment becoming partially disordered, loop 11 that contains the 'switch II' motif adopts an ordered helical turn (Fig. 2e), whilst the MT-interacting helix α4 is extended (Fig. 2f). Overall, these observations indicate that, bar the presence of the bound nucleotide, the MT-bound structure of ADP-Kif5b[MoNeXXX] resembles more the nucleotide-free conformation of Kif5b in complex with a tubulin dimer and a DARPin (PDB code 4lnu)[29] than the MT-free ADP-bound structure of Kif5b (Supplementary Fig. 10b).

The nucleotide-free (apo) conformation of kinesin-1 represents a 'strong' binding state[48], with enhanced MT contacts formed by the ordering of the α4 helix extension and loop 11 region[29]. Structural superposition upon the P-loop/α2a helix that holds ADP reveals that our structure and the tubulin dimer-bound motor in the apo state exhibit low RMSD values globally (Fig. 2g), whilst MT-free $Mg^{2+}$-ADP-bound Kif5b is structurally divergent on one side (Fig. 2h). If analyzed in terms of the kinesin subdomain scheme used in previous studies[26,29] (Fig. 2i), the whole P-loop subdomain stays relatively unchanged while the switch I/II and MT-binding subdomains undergo significant conformational rearrangements. This results in a novel network of interactions between conserved residues of the extended helix α4/loop 11 region and loop 7/loop 9 of the switch I/II subdomain. For example, E236 of switch II (in loop 11), which bonded T87 of the P-loop in the MT-free $Mg^{2+}$-ADP structure locking the P-loop and loop 11 together in a manner crucial to stabilizing the $Mg^{2+}$-ADP state[49] (Fig. 2d), moves away from the P-loop upon MT-binding, coming into close proximity with R203 of switch I (in loop 9) (Fig. 2e). In our structure, loop 7/loop 9 and helix α4-proximal regions of the core β-sheet in the switch I/II subdomain (β strands 4–7) are drawn towards helix α4 and the MT-binding subdomain, while regions of the core β-sheet in the P-loop subdomain (β-strands 1, 3 and 8) remain relatively static (Fig. 2f). This causes 'twisting' of the core β-sheet that has been suggested to happen upon ADP release[29]. However, we show here that this occurs upon MT-binding and $Mg^{2+}$ release prior to ADP exit.

## Structural basis for a two-step MT-stimulated ADP release mechanism

MT-stimulated ADP release has been proposed to be biphasic, with the first phase being inhibited by $Mg^{2+}$[18,30,50]. This is consistent with an ADP-bound transition state lacking $Mg^{2+}$ and representing the weak-to-strong MT-binding state. This has been suggested on the basis of FRET measurements to anticipate and accelerate ADP release[25]. Our cryo-EM sample preparation conditions have a >1000-fold molar excess of $Mg^{2+}$ and ADP compared to the motor. Yet, while $Mg^{2+}$ density is absent, ADP is clearly present, indicating that MT binding drastically reduces motor

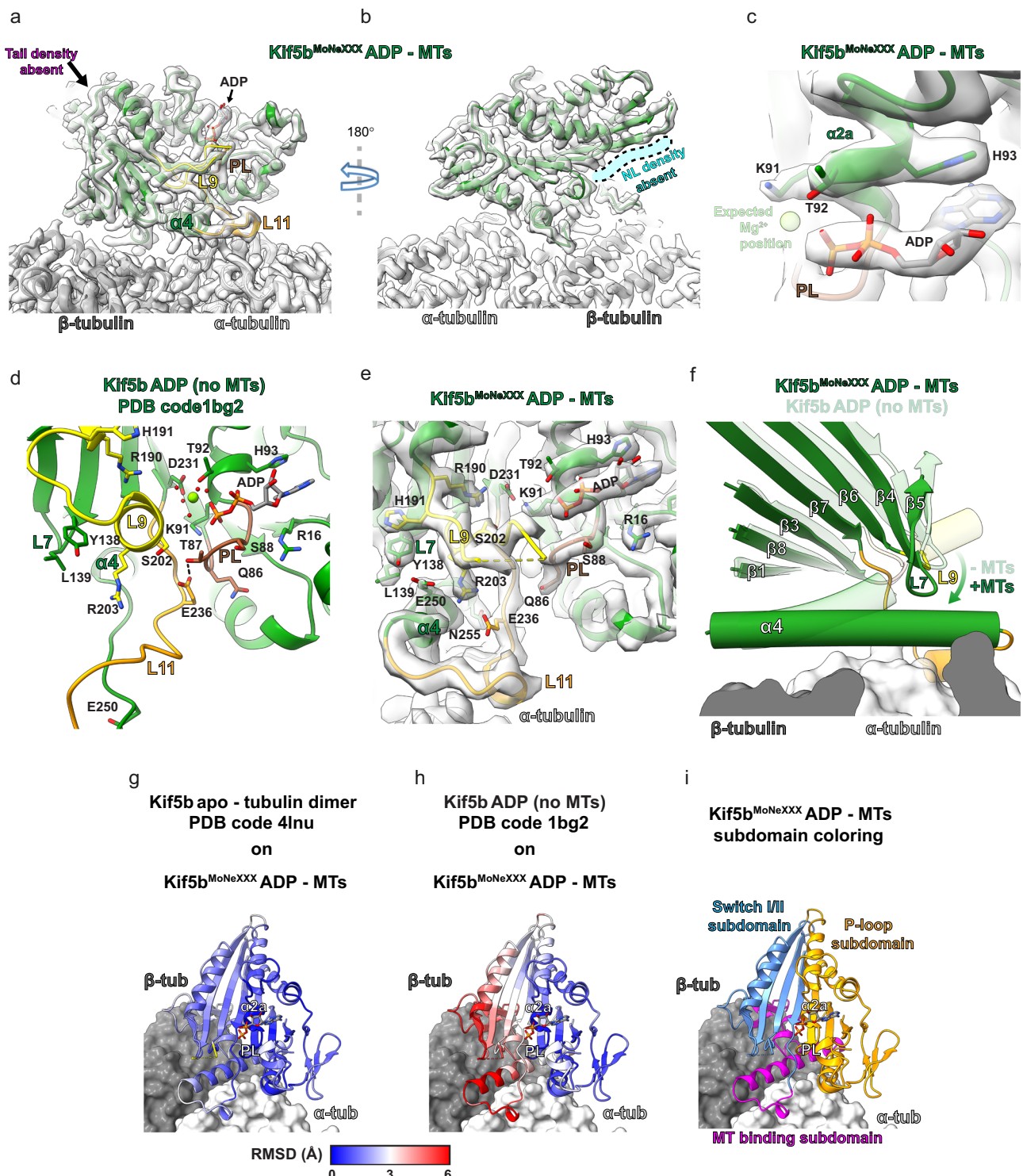

**Fig. 2 | Kif5b motor domain binding to MTs causes dissociation of the Mg²⁺ ion from ADP and adoption of an apo-like conformation.** **a**, **b** Two views of the MT-bound Kif5b^MoNeXXX ADP-state reconstruction with the model fitted into density (gray transparent). In (**b**), the position where density for a docked neck linker (NL) could be expected (absent here) is indicated in semi-transparent cyan color. **c** View of the nucleotide pocket of MT-associated Kif5b^MoNeXXX-ADP, highlighting the lack of cryo-EM density at the Mg²⁺ position (semi-transparent lime). **d**–**f** Overviews of the nucleotide pocket and switch-motif containing loops 9 (L9) and 11 (L11), with side chains for key mechanistic residues shown for **d** the crystal structure of Kif5b-ADP motor domain without MTs (PDB code 1bg2) and **e** our Kif5b^MoNeXXX-ADP structure in the presence of MTs (cryo-EM density in transparent gray). In (**f**), a superimposition highlighting helix α4's extension, a twist of the overlying core β-sheet, and

concomitant movement of overlying loop 7 (L7) and loop 9 (L9) upon transition from the Kif5b-ADP state without MTs (PDB code 1bg2, semi-transparent model) to our MT-bound Kif5b^MoNeXXX ADP-state (opaque model). Helices are shown as tubes, and αβ-tubulin is shown as a gray surface representation. **g**, **h** (**c**) The crystal structures of the Kif5b motor domain without nucleotide (apo) bound to αβ tubulin dimer (PDB code 4lnu) or (**d**) of Kif5b-ADP without MTs (PDB code 1bg2) are superimposed on our Kif5b^MoNeXXX ADP-state, and calculated RMSDs are shown. The superimpositions in (**f**–**h**) use the nucleotide-holding P-loop and helix α2a elements for alignment. **i** Kif5b^MoNeXXX ADP-state colored according to the kinesin motor domain subdomain scheme[26,29]. Cryo-EM density in all panels was filtered using DeepEMhancer[62], apart from (**c**), which was sharpened by local resolution in Relion.

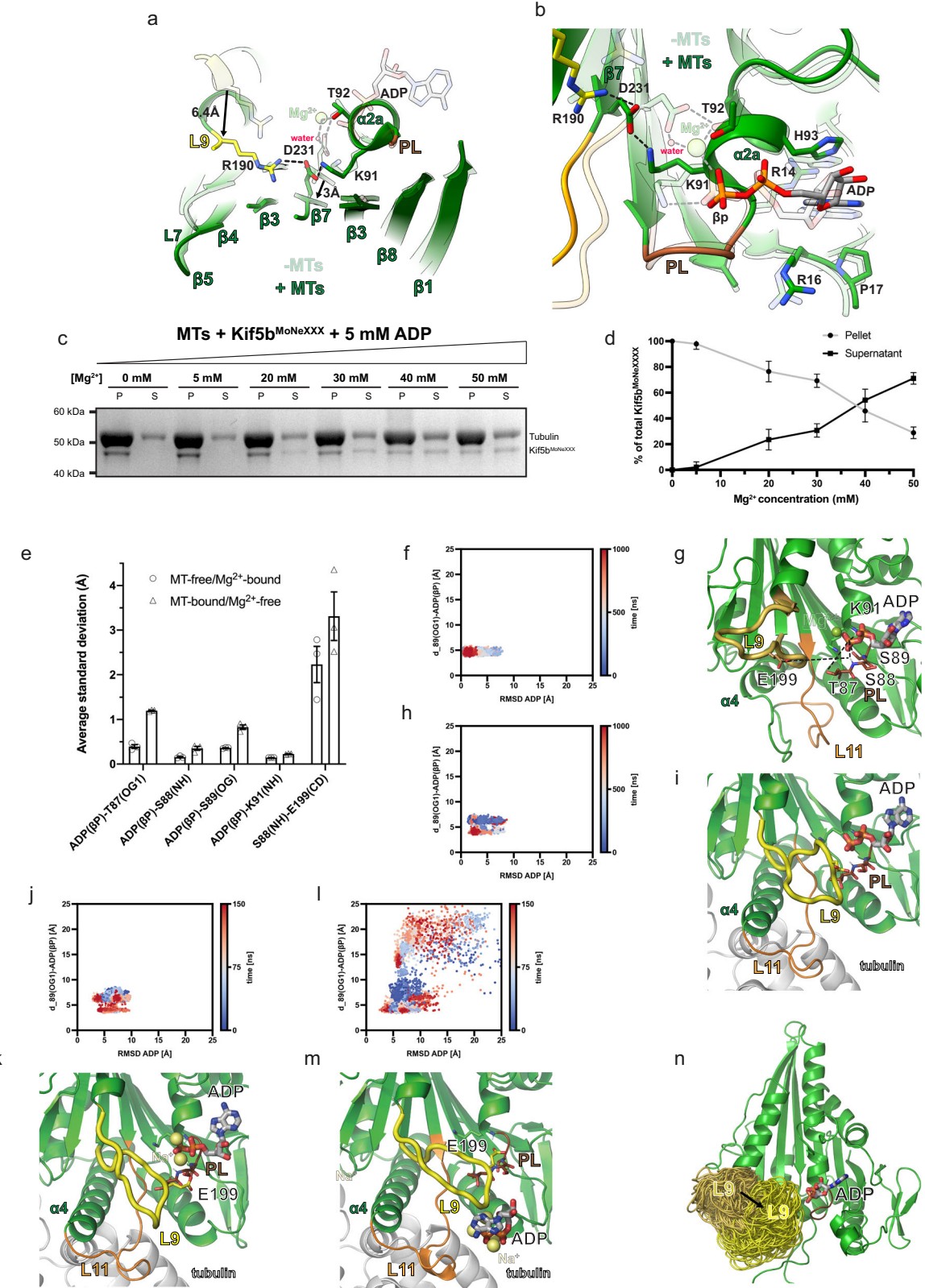

affinity for the ion, but has, together with Mg²⁺ release, a comparatively less pronounced effect on ADP. Retention of the latter, owing to its very high concentration, allowed us to analyze the conformational changes that impact ADP affinity following MT binding and Mg²⁺ release.

The MT-induced structural changes discussed in the previous section induce significant displacements of key Mg²⁺-coordinating side

chains of loop 9 and β7 relative to ADP, the ADP-binding P-loop, and the helix α2a region in the P-loop subdomain that are not compatible with Mg²⁺ retention. In particular, prior to Kif5b interaction with the MT, conserved D231 on β7 indirectly coordinates Mg²⁺ via a water molecule and the hydroxyl group on helix α2a's T92 (Fig. 3a). Upon MT binding, however, the Cα atoms of D231 are displaced by ~3 Å relative to the P-loop and helix α2a, such that these interactions are broken

**Fig. 3 | A two-step mechanism for MT-stimulated ADP release in Kif5b is supported by cryo-EM and MD simulations. a, b** Superimpositions of Kif5b-ADP without MTs (PDB code 1bg2, semi-transparent model) with our Kif5b[MoNeXXX] ADP-state (opaque model) highlighting the proposed **a** Mg$^{2+}$-release and **b** ADP destabilization mechanisms. Hydrogen bonds in the Kif5b-ADP state without and with MTs are shown as gray and black dashed lines, respectively. In (**a**), the movement of R190 and D231 α-carbon atoms is illustrated with black arrows. **c** Representative co-sedimentation assay of MT and Kif5b[MoNeXXX] co-incubated with increasing amounts of MgCl$_2$ as indicated. Co-sedimentation was performed with taxol-stabilized MTs at 2 μM (tubulin dimer) and 0.5 μM Kif5b[MoNeXXX]. P = Pellet, S = Supernatant. Molecular weights are indicated. **d** Densitometric quantification of Coomassie-stained bands in the SDS-PAGE analysis of co-sedimentation assays (as in **c**), showing the percentage of total Kif5b[MoNeXXX] in pellet vs. supernatant fractions. $n = 4$ independent experiments for each condition. Data are presented as mean values ± SD. Source data are provided as a Source Data file. **e** Average standard deviation for the atomic distances used to monitor ADP stability within the binding pocket during three independent 1 μs MD replicas. Error bars are s.e.m. values. Source data are provided as a Source Data file. The MT-free/Mg$^{2+}$-bound system is Kif5b-ADP without MTs (PDB code 1bg2). The MT-bound/Mg$^{2+}$-free system is modeled based on our Kif5b[MoNeXXX] ADP-state. **f** 2D distribution of two of the reaction coordinates: RMSD(ADP) and distance between its phosphorus Pβ and S98, monitored in the MD runs for the MT-free/Mg$^{2+}$-bound system and **g** representative snapshot. Atom pair distances reported in (**e**) are shown as broken black lines. **h, i** like (**f, g**) for the MT-bound/Mg$^{2+}$-free system. Compared to the MT-free/Mg$^{2+}$-bound system, here loop 9 moves closer to the active site. **j, k** like (**h, i**) for Level 2 replicas. **l, m** like (**h, i**) for Level 3 replicas, leading to ADP dissociation. **n** Cartoon representation highlighting the loop 9 dynamics in the MT-free/Mg$^{2+}$-bound system (dark yellow) and in the MT-bound/Mg$^{2+}$-free system (yellow). In the latter system, there is a shift toward the active site that enables ADP displacement, as shown in (**m**). Source data for panels (**f, h, j, l**) are provided as a Source Data file.

with D231 forming instead a new hydrogen bonding network with conserved R190 on loop 9 (also enabled by its Cα atom moving ~6.5 Å relative to the P-loop) and conserved K91 on α2a (Fig. 3a). Moreover, while interactions between ADP and the backbone of the P-loop and helix 2a as well as with the side chains of R14, R16, P17, and H93 are maintained, thus conferring residual stability to the nucleotide in its binding pocket, the β-phosphate-coordinating K91 side chain adopts a new rotamer that is H-bonded to D231 (Fig. 3b). The reorientation of K91, alongside the loss of β-phosphate-coordination by Mg$^{2+}$ and its associated water network is therefore expected to lead to a significant loss of ADP affinity driving its release under physiological conditions.

Given that the conformational changes driven by MT binding leading to a 'strong' MT-bound state are incompatible with Mg$^{2+}$ remaining associated with ADP in the nucleotide pocket, we predicted that in the presence of ADP, increasing Mg$^{2+}$ concentrations would reduce MT affinity by preventing the conformational changes that lead to the 'strong' MT-bound state. To assess this, we performed MT co-sedimentation assays with Kif5b[MoNeXXX] in the presence of ADP and increasing concentrations of Mg$^{2+}$. As expected, MTs pellet at high centrifugal forces, while in the absence of MTs Kif5b[MoNeXXX] remains chiefly in the supernatant (Supplementary Fig. 11). In the presence of a molar excess of MTs, we find that at [Mg$^{2+}$] ≲ 10 mM all detectable Kif5b[MoNeXXX] binds and co-pellets with the filaments (Fig. 3c). However, as [Mg$^{2+}$] rise above ~10 mM, Kif5b[MoNeXXX] progressively moves from the pellet to the supernatant, indicating that MT binding is reduced in a [Mg$^{2+}$]-dependent manner (Fig. 3c, d). We could not test Mg$^{2+}$ concentrations above 50 mM due to significant MT destabilization; however, a large molar excess is clearly needed to decrease Kif5b[MoNeXXX] MT association. This is consistent with the notion that MTs increase kinesin-1 ATPase rates by ~1000 fold, by stimulating the rate-limiting Mg$^{2+}$ and subsequent ADP release steps[16,50]. Overall, considering the apo-like conformation of our MT-bound and ADP-bound Kif5b[MoNeXXX] structure, we believe to have captured kinesin-1 in the pre-ADP release 'strong' MT-binding state, induced by Mg$^{2+}$ dissociation.

To our knowledge, an Mg$^{2+}$-free ADP-bound apo-like conformation of Kif5b on MTs as described here has only been observed previously in the atypical kinesin Kif14[51]. When the motor domain of the latter (PDB code 6wwm) was superimposed onto the motor domain in our ADP-bound Kif5b[MoNeXXX] model, the structures showed high similarity apart from the divergent loops 2, 3 and 8 and 9, which was similar at its base but more ordered at its apex in Kif14 (Supplementary Fig. 12a). Furthermore, a similar 'apo-like' network of key conserved residues was seen (Supplementary Fig. 12b), including the interactions between R591, D638 and K488 (equivalent to Kif5b R190, D231 and K91 respectively) which we regard as critical for the Mg$^{2+}$ and ADP release mechanisms (Fig. 3a, b). Kif14 also exhibits an apo-like ADP-bound conformation lacking Mg$^{2+}$ in the absence of MTs[52], except for loop 11 being more disordered and loop 9 being more disordered at its apex

(Supplementary Fig. 12c). Compared to other kinesins, Kif14 has an uncommonly high affinity for MTs in the presence of ADP[52], which can be explained by the lack of Mg$^{2+}$ coordination and the adoption of this apo-like conformation, including an extended/stabilized helix α4 MT-interacting element, even in the absence of MTs. The extension and orientation of helix α4 in Kif14 in the absence of MTs coincides with a core-β twist and movement of the overlying loop 9, leading to the R591 and D638 interaction (R190 and D231 in Kif5b) precluding Mg$^{2+}$ binding (Supplementary Fig. 12c). An apo-like conformation in solution explains the unusually high basal ATPase rate of Kif14[52], as ADP release is accelerated by the lack of Mg$^{2+}$ coordination. Interestingly, however, MTs still stimulate ATPase in this motor, but only by a further 3-fold as opposed to more than ~104-fold in kinesin-1[16,52]. When comparing MT-bound[51] and MT-free[52] ADP-Kif14 structures, this can likely be explained by the reorientation of K488 side chain upon MT interaction to interact with D638 instead of ADP's β-phosphate, further reducing ADP affinity and accelerating its release (Supplementary Fig. 12b, c).

## ADP dissociates from the MT-bound/Mg$^{2+}$-free Kif5 motor in Molecular Dynamics simulations

To further understand the structural basis of ADP dissociation from the Kif5 motor, we performed all-atom Molecular Dynamics (MD) simulations of the MT-bound/Mg$^{2+}$-free ADP-Kif5b[MoNeXXX] complex and of the MT-free/Mg$^{2+}$-loaded complex (PDB code 1bg2) for comparison. Each system was simulated in three replicas for a total time of 3 μs. To assess the structural stability of bound ADP, we monitored the variation during the simulations of a set of distances between its β-phosphorus atom (βP) and P-loop residues (T87, S88, S89, K91) as well as the distance between S88 and E199. These exhibited larger fluctuations in the MT-bound complex compared to the MT-free one, pointing to a lower ADP stability in the former (Fig. 3e). The increase in distance fluctuations is correlated with a shift of the RMSD(ADP) distribution towards larger values and correspondingly, with its higher average (3.2 Å and 1.7 Å for the MT-bound/Mg$^{2+}$-free and the MT-free/Mg$^{2+}$-loaded systems, respectively) (Fig. 3f, g). Representative snapshots of these simulations are shown in Fig. 3h, i.

To test the hypothesis that ADP might dissociate in the MT-bound system on a longer time scale, we applied an adaptive iterative strategy yielding a total of 9.75 μs of simulation time (Supplementary Fig. 13). In this protocol, we extracted from the initial MD run (we refer to this as Level 1), two time points exhibiting high distance and RMSD(ADP), as these likely reflect relative energy maxima. These were then used to restart the simulations generating a new swarm of ten (5 × 2) parallel replicas (Level 2) that are characterized by an increase in RMSD(ADP) (Fig. 3j and Supplementary Fig. 14, cartoon snapshot in Fig. 3k). Next, we ran an additional set of 35 new replicas (Level 3) starting from seven high-RMSD(ADP) snapshots from the Level 2 dataset. These display a significant further increase in RMSD(ADP) (Fig. 3l and Supplementary

Fig. 14, cartoon snapshot in Fig. 3m) and in four of these, even complete ADP detachment with its release in the bulk (see Supplementary Video 1). The mechanistic basis for this can be linked to two main structural differences that accompany the transition from the MT-free to MT-bound state: (i) the extension of the MT-interacting α4 helix and (ii) the loss of the helical portion of loop 9 that becomes partially disordered. MD shows that these have a significant impact on the dynamics of the ADP site as the longer α4 helix not only sterically destabilizes loop 9 but also contributes to its relocation in proximity of the catalytic site (Fig. 3i, k, m, n). Here, loop 9, only partly stabilized by compensatory $Na^+$ ions (we observe three on average in the simulations) that intervene to mitigate the loss of coordinating $Mg^{2+}$ discussed in the previous section, directly displaces ADP with E199 transiently occupying the phosphate pocket (Fig. 3m). Overall, MT-binding drives a set of global and active site-specific structural changes that trigger the loss of $Mg^{2+}$ culminating in ADP destabilization and its eventual detachment mediated by loop 9 dynamics.

### Tail association, with or without zippering of the distal head, does not affect the ADP-bound conformation of the MT-bound head

To assess the structural impact of the tail on the ADP-bound conformation, we superimposed the one-headed ADP-bound Kif5b[MoNeIAK] model onto ADP-bound Kif5b[MoNeXXX]. The models are almost identical in all regions (RMSD = 0.3 Å), demonstrating that tail association has little effect on the ADP-bound structure of the MT-bound head (Fig. 4a). When superimposing the one-headed and two-headed ADP-bound Kif5b[MoNeIAK] models on the MT-associated head (Fig. 4b), again, negligible structural differences were observed (RMSD = 0.2 Å). Despite the presence of the tail, the MT-bound head of Kif5b[MoNeIAK] adopts the same apo-like ADP-bound and $Mg^{2+}$-less conformation seen in Kif5b[MoNeXXX], with an extended helix α4 and ordered loop 11 in the MT-binding subdomain and an unfurled partially disordered loop 9 (Fig. 4c). Furthermore, the bonding network between loop 9's R190, β7's D231 and helix α2a's K91 described for Kif5b[MoNeXXX] (Fig. 3b) is also present in the MT-associated head of one- (Fig. 4d) and two-headed states of Kif5b[MoNeIAK] (Fig. 4e). This strongly suggests that the IAK-motif tail, irrespective of the presence of the zipped distal head, has little effect on the MT-stimulated $Mg^{2+}$ or ADP release mechanisms of Kif5b.

Supporting the role of MT binding in the conformational change observed in the MT-associated head leading to $Mg^{2+}$ release, density (although at lower resolution and low-pass filtered to 6 Å for reliable interpretation) in the distal head of Kif5b[MoNeIAK] is most consistent with loop 9 transitioning to a short helical segment, helix α4 being short and partly disordered, loop 11 being mainly disordered and $Mg^{2+}$-ADP remaining associated (Fig. 4f, g and Supplementary Fig. 15). This strongly resembles the $Mg^{2+}$-ADP-bound conformation observed previously for tailless Kif5b motor domain without MTs[47].

### Tail association, with or without zippering of the distal head, does not affect the ATP-hydrolysis-competent conformation of the MT-bound head

To assess the structural impact of the tail on the 'ATP-like' conformation, we superimposed the one-headed AMPPNP-bound Kif5b[MoNeIAK] model onto the X-ray crystallographic model of ADP-AlF$_4$-bound Kif5b in complex with a tubulin dimer and a DARPin (PDB code 4hna)[15] (Fig. 5a). The models superimposed very well in all regions of the motor domain (RMSD = 0.5 Å), demonstrating tail association has little effect on the ATP-like state of the MT-associated head (Fig. 5a). Upon superimposing the one-headed and two-headed AMPPNP-bound Kif5b[MoNeIAK] models on the MT-associated head (Fig. 5b), again a high level of structural similarity was observed (RMSD = 0.2 Å). This indicates that even the additional presence of the 2nd-head zipped via the IAK motif has a negligible or no structural effect on the MT-bound head. Despite the presence of the tail or the zipped 2nd head therefore,

the MT-bound head has an extended helix α4 and ordered loop 11 in the MT-binding subdomain, associated with a 'strong' MT-bound state (Fig. 5a–c). The helical turn of switch-II containing loop 11 contacts a fully ordered switch-I-containing Loop 9, closing AMPPNP and coordinated $Mg^{2+}$ between these elements and the P-loop (Fig. 5c). The crystallographic structure of a kinesin-5 (Eg5)[53] demonstrated that this 'closed' conformation allows ATP's γ-phosphate to be stabilized by $Mg^{2+}$ and residues equivalent in Kif5b to S201 and S202 of switch I in loop 9, whilst preparing it for nucleophilic attack via waters arranged by S202 and R203 of switch I (in loop 9) and G234 and E236 of switch II (in loop 11) (Fig. 5d, e & Supplementary Fig. 16).

In the two-headed AMPPNP-Kif5b[MoNeIAK] state, the 2nd head was resolved at lower resolution (~4–6 Å), particularly at regions distal to the tail and tip of the motor domain (Supplementary Fig. 4c), such that low-pass filtering its density to 6 Å was appropriate to examine its conformation. Nonetheless, this revealed that while $Mg^{2+}$-AMPPNP occupied the nucleotide pocket, helix α4 was not extended, and loops 9 and 11 were not well ordered, such that the 2nd head was not in a nucleotide-hydrolysis-competent conformation (Fig. 5f, g). This finding is consistent with the requirement of MT binding for ordering of these secondary structure elements and the adoption of a fully hydrolysis-competent conformation[15,17,18,20].

### MT-bound Kif5b[MoNeIAK] undocks its neck linker in the presence of ADP regardless of tail binding

In the presence of MTs, closing of the nucleotide pocket in response to ATP analogs induces conformational changes in kinesin motor domains, leading to extension of helix α6 and docking of the preceding neck linker[20,24,54]. After Pi release following hydrolysis, the neck linker then undocks and becomes disordered, along with the C-terminus of helix α6, through both $Mg^{2+}$-ADP-bound and apo stages of the ATPase cycle[19,20]. In contrast, in the absence of MTs, neck linker docking is not coupled with the nucleotide state[55]. In the crystallographic structure of the ADP-bound *Drosophila* kinesin-1 motor dimer in complex with an IAK motif peptide, the neck linkers in both heads are docked, leading into the coiled-coil[35]. From this structure, the tail was suggested to prevent neck linker undocking, and that this mechanism inhibits MT-stimulated ADP release (double-lockdown mechanism).

In the presence of AMPPNP, the MT-associated head of both single and two-headed Kif5b[MoNeIAK] states exhibited an extended helix α6 and docked neck linker (Fig. 6a, b) coincident with closing of the nucleotide pocket around the ATP analog (Fig. 5c). In the two-headed state, helix α6 was extended, and the neck linker was docked in the 2nd head as well, such that a portion of the dimeric coiled-coil could be resolved (Fig. 6b and Supplementary Fig. 17a). In contrast to a processive dimer however, the 2nd head is not thrust forward onto the next αβ-tubulin binding site towards the MT plus end but is instead locked in a restricted orientation against the 1st head by the tail (Fig. 6b & Supplementary Fig. 8a). In contrast, in the presence of ADP, the MT-associated head of Kif5b[MoNeIAK] clearly exhibited a disordered (and undocked) neck linker and shortened helix α6 in both single- and two-headed states (Fig. 6c, d). In contrast to the MT-associated head, the 2nd head in the two-headed state has an extended helix α6 and docked neck linker (Supplementary Fig. 17b, c). Consistent with the lack of neck linker docking in the MT-associated head, we did not observe ordering of CC1 (Fig. 6d and Supplementary Fig. 17b). As the two-headed class had lower occupancy in the presence of ADP compared to AMPPNP (Fig. 1i), it is likely that the stabilization of a docked neck linker and CC1 (Fig. 6b) supports a two-headed complex including a more ordered tail (Fig. 1k, m). Regardless, these reconstructions therefore demonstrate that nucleotide-dependent neck linker undocking still occurs in the MT-associated head, regardless of the presence of the autoinhibitory tail with or without a zippered 2nd head.

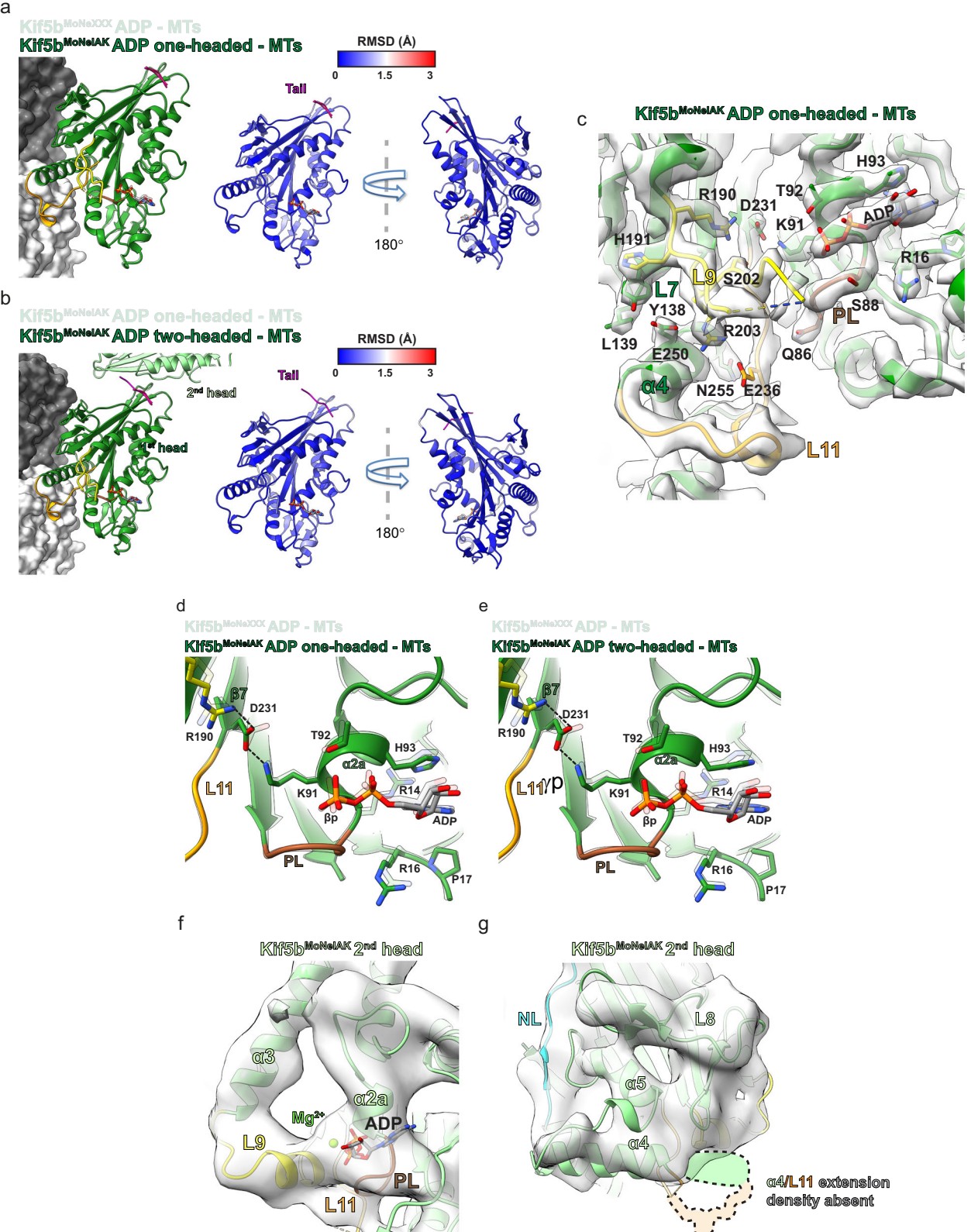

## Discussion

MT-induced stimulation of ATP hydrolysis and cargo-dependent relief of autoinhibition are key aspects of kinesin-1's mechanochemistry as they contribute to preventing futile mechanochemical cycles in the absence of productive motor runs. The rate-limiting step in the kinesin-1 ATPase cycle is ADP release[16], and various studies have shown that this follows a two-step sequential mechanism whereby $Mg^{2+}$

release precedes and accelerates that of ADP, with the weak-to-strong MT-binding state transition anticipating the ADP release step[18,25,30,50].

Our structure of ADP-loaded Kif5b bound to its MT track captured the 'strong' MT-bound state of the motor after $Mg^{2+}$ dissociation and before ADP release, allowing us to describe a structural model for the two-step mechanism of MT-stimulated ADP release (Fig. 7). Transition to the 'strong' binding state coincides with the

**Fig. 4 | Tail or 2nd head association has no effect on the ADP-bound conformation of the MT-associated Kif5b motor domain. a** ADP-bound motor domains of the Kif5b[MoNeIAK] and Kif5b[MoNeXXX] (semi-transparent) MT-associated models were superimposed (left). RMSDs were calculated and displayed on the Kif5b[MoNeIAK] model (right), with the Kif5b[MoNeIAK] tail shown in magenta. **b** MT-associated ADP-bound motor domains of the Kif5b[MoNeIAK] one (semi-transparent) and two-headed models were superimposed (left). RMSDs were calculated and displayed on the Kif5b[MoNeIAK] two-headed model (right) with the Kif5b[MoNeIAK] tail shown in magenta. **c** Overview of the nucleotide-pocket and switch-motif containing loops 9 and 11 (L9 and L11) in the ADP-bound Kif5b[MoNeIAK] one-headed model, with side chains for key mechanistic residues shown. Cryo-EM density is shown as

semi-transparent gray and was filtered using DeepEMhancer[62]. **d**, **e** Superimposition of the ADP-bound MT-associated motor domains of Kif5b[MoNeXXX] (semi-transparent) and **d** the Kif5b[MoNeIAK] one-headed model, **e** the Kif5b[MoNeIAK] two-headed model. A view of the nucleotide-binding site is shown, alongside key side chains involved in $Mg^{2+}$ and ADP release. **f**, **g** Views of the 2nd (not MT-associated) head in the Kif5b[MoNeIAK] two-headed model in the presence of ADP. Cryo-EM density (semi-transparent) was low-pass filtered to 6 Å resolution for resolution-appropriate visualization. The position where density for an extended helix α4 and ordered L11 could be expected (but is absent here) is indicated in semi-transparent colors with dashed black outlines.

ordering of the MT-interacting helix α4 and loop 11 elements prior to ADP release. This produces a new set of interactions with the core β-sheet and associated loop 7 and loop 9, causing a 'twist' of the β-sheet and displacement of these loops relative to the nucleotide-holding P-loop/helix α2a. As a result, interactions between the conserved D231 of the core β-sheet (a switch-II motif residue), T92 (helix α2a), and the water 'cap' holding the $Mg^{2+}$ ion are broken, leading to its release (step 1). Concomitantly, D231 forms a new hydrogen bonding network with R190 on the displaced loop 9 and with K91 on helix α2a, such that neither $Mg^{2+}$ nor K91 remains bound to the β-phosphate of ADP, suggesting a mechanism for MT-dependent and $Mg^{2+}$-loss-triggered stimulation of ADP release (step 2). A recent magic-angle-spinning NMR study of nucleotide-free Kif5b bound to MTs also shows an interaction between K91 and D231, as in our ADP-bound structure[56]. The importance of conserved D231 and R190 as part of the '$Mg^{2+}$-stabilizer' network has previously been noted, and replacement of these residues drastically reduced the ATPase rate whilst increasing ADP release rates in kinesins[29,30]. The key role of $Mg^{2+}$ in ADP retention is also clear from the observation that mutation of the P-loop $Mg^{2+}$-coordinating residue T92 results in a drastic loss of ADP affinity[49]. Our observation that K91 additionally no longer supports the ADP β-phosphate suggests this as an auxiliary factor in reducing ADP affinity in response to MT binding. K91 is a highly conserved residue in the P-loop Walker A motif (GxxxxG**K**(S/T)) that spans P-loop and helix α2a structural elements, and a substitution at the site was recently identified as causing kyphomelic dysplasia[57].

Another finding of the present study is that the IAK motif-containing tail, even when zippering the second head, has no structural effects on the MT-bound motor in the presence of either ADP or AMPPNP. Therefore, any influence of the IAK tail on kinesin-1's mechanochemistry is unlikely to be mediated by steric or allosteric effects on the nucleotide-binding site affecting nucleotide binding, release, or hydrolysis. This contrasts with a previous ~8 Å cryo-EM study of a kinesin-1 motor domain without nucleotide, which found tail peptide density alongside the switch I-motif containing loop 9, which was suggested to inhibit ADP release by preventing conformational changes around the MT-binding region and nucleotide pocket upon MT binding[34]. However, as this sample was chemically cross-linked, sample preparation artefacts cannot be ruled out. The location of the bound tail and the mode of zippering of the second head observed in our study are akin to those seen in the X-ray crystallographic structure of the MT-free ADP-bound autoinhibited *Drosophila* kinesin-1 dimer[35]. In the latter structure, CC1 was ordered with the neck linker docked in both heads prompting the authors to suggest a 'double-lockdown autoinhibition mechanism', whereby the tail locks the two heads in a restricted orientation (lock 1) that, in turn, prevents neck linker undocking (lock 2) such that ADP release is inhibited. However, we unambiguously show here that in the presence of ADP, the MT-associated motor domain undocks its neck linker regardless of the presence of the bound tail or of the distal motor head zippered to the first. This implies that the 'double-lockdown' mechanism of autoinhibition needs to be refined.

What role does tail binding then play in kinesin-1 autoinhibition? Full-length kinesin-1 with or without KLCs is autoinhibited in a manner that strongly suppresses MT association[33,39,40], and according to recent models based on low-resolution negative stain EM and crosslinking mass-spectrometry, autoinhibited full-length kinesin-1 folds such that regions of the stalk obscure a MT-binding surface of at least one of the motor domains[36–38]. In these models, one of the two IAK motif containing tails in the dimer folds back to associate with and zip up the two motor domains (as in the current study), thanks to an 'elbow' that allows flexibility in the stalk. Although kinesin-1 appears to have a built-in tendency to fold back on itself[37], the presence of the IAK motif containing tail is required for full inhibition of ATPase activity, ADP release, and motility[31,33,38,40]. Taken together, in full-length kinesin-1, a key function of the IAK motif containing tail appears that of acting as a 'safety belt', ensuring a stable compacted fold, thereby indirectly preventing MT landings due to regions of the stalk sterically blocking the MT-binding interface. However, although MT landing events of cargo-free full-length kinesin-1 are minimal, a basal level remains, and when MT-associated, these motors display reduced motility with frequent pausing and longer association/run durations[37,40]. Furthermore, truncated dimeric kinesin-1 constructs lacking the tail, and in some cases most of the coiled-coil, associate strongly with MTs but are stalled and display reduced MT-stimulated ATPase and ADP release rates in the presence of exogenous tail peptide[41,42,44]. Therefore, this represents an additional autoinhibitory mechanism mediated by the IAK-containing tail independent of simple reduction of MT-binding via steric inhibition in the folded conformation. The structures presented here demonstrate that this does not arise from any direct influence of the tail on the conformational transitions of the MT-associated head and suggest that initial MT-stimulated ADP release, ATP binding, or hydrolysis steps are unaffected. However, our structures would indicate that upon ATP binding to the MT-associated head, the distal one is not thrust forward by neck linker docking but is instead restricted by the tail in a new orientation dictated by the two-headed complex. Consequently, the second head is not presented to the next binding site on the MT track, thus preventing MT-stimulated ADP release and ultimately processive motility. In support of this hypothesis, an initial transient of MT-stimulated ADP release occurs in the presence of the tail before further ADP release is prevented, with a similar effect found when chemically crosslinking the heads together in a tailless construct[35]. This mechanism can, in principle, also account for tail-dependent stalling of kinesin-1 on MTs in a strongly associated state[37,39–42,44]. As only one of the two tails of kinesin-1 associates with the heads[35] and promotes autoinhibition[32], it is possible that inter-molecular interactions with other kinesin-1 molecules results in zippering, leading to cross-motor inhibition, resulting in the characteristic tail-dependent pausing events. Autoinhibition in kinesin-1, therefore, appears to be composed of multiple mechanistic layers, including those dependent on steric hindrance of MT association by regions of the stalk and those dependent on the reduction of processive stepping behavior.

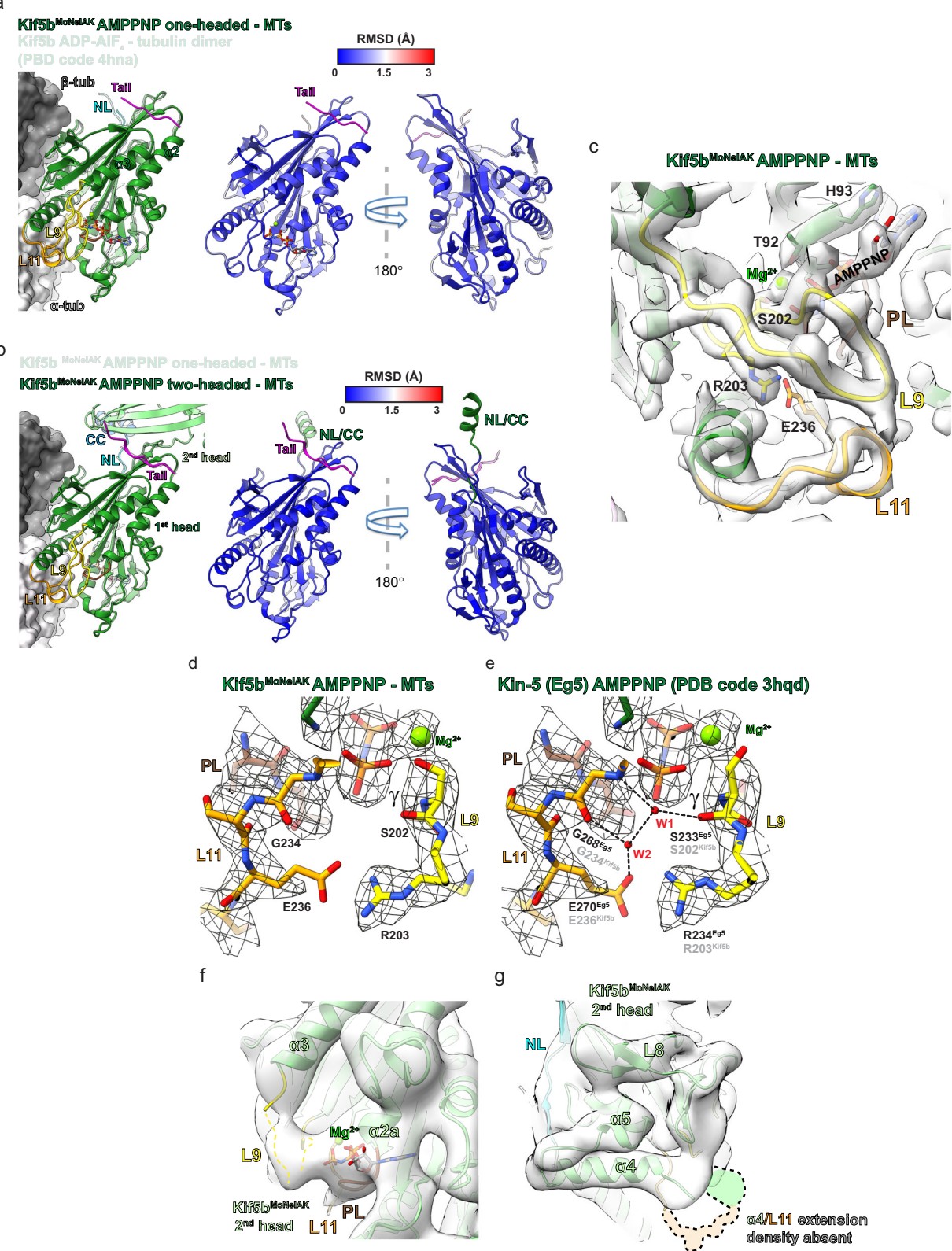

## Methods

### Protein expression and purification

A codon-optimised DNA sequence of human Kif5b (Uniprot entry P33176) encoding the motor domain, neck linker, and CC1 (residues 1–357) fused via a flexible (Thr-Gly-Ser)$_9$ linker to the C-terminal tail region encompassing the autoinhibitory IAK motif (residues 912–935) was purchased from Genscript and subcloned between the NdeI/XhoI sites of a pET28 vector (Novagen). We refer to this chimeric protein as Kif5b$^{MoNeIAK}$ for Kif5b Motor-Neck-IAK. A second construct was also synthesized that replaces the IAK motif region (919-QIAKPIR-925) with the flexible palindromic sequence (TGSTSGT). We refer to this construct as Kif5b$^{MoNeXXX}$.

Kif5b$^{MoNeIAK}$ and Kif5b$^{MoNeXXX}$ were expressed in the *E.coli* BL21(DE3) strain. Briefly, single colonies were picked and grown at 21 °C overnight

**Fig. 5 | Tail or 2nd head association has no effect on the ATP-like conformation of the MT-associated Kif5b motor domain. a** Motor domains of the Kif5b[MoNeIAK] AMPPNP and MT-bound and ADP-AlF$_4$ and tubulin-bound (semi-transparent, PDB code 4hna) models were superimposed (left). RMSDs were calculated and displayed on the Kif5b[MoNeIAK] model (right), with the Kif5b[MoNeIAK] tail shown in magenta. **b** MT-associated AMPPNP-bound motor domains of the Kif5b[MoNeIAK] one (semi-transparent) and two-headed models were superimposed (left). RMSDs were calculated and displayed on the Kif5b[MoNeIAK] two-headed model (right) with the Kif5b[MoNeIAK] tail shown in magenta and the NL-CC region shown in green. **c** Overview of the nucleotide-pocket and switch-motif containing loops 9 and 11 (L9 and L11) in the AMPPNP-bound Kif5b[MoNeIAK] one-headed model, with side chains for key conserved residues shown. Cryo-EM density is shown as semi-transparent gray and was filtered with DeepEMhancer[62]. **d, e** Close-up view of residues involved in ATP

hydrolysis, comparing **d** our MT-associated AMPPNP-bound Kif5b[MoNeIAK] motor domain (left) and **e** Eg5 motor domain (PDB code 3hqd). Both models are fitted into density (local resolution sharpened in Relion) for the MT-associated motor domain of AMPPNP-bound Kif5b[MoNeIAK] (mesh). The high resolution in the Eg5 motor domain crystal structure allows display of the two water molecules and associated bonding network (black dashed lines) that are involved in nucleophilic attack on the γ-phosphate[53]. In the right-hand panel, Eg5 residue numbering is shown in black and Kif5b residue numbering in gray. **f, g** Views of the 2nd (not MT-associated) head in the Kif5b[MoNeIAK] two-headed model in the presence of AMPPNP. Cryo-EM density (semi-transparent) was low-pass filtered to 6 Å resolution for resolution-appropriate visualization. The position where density for an extended helix α4 and ordered L11 could be expected (but is absent here) is indicated in semi-transparent colors with dashed black outlines.

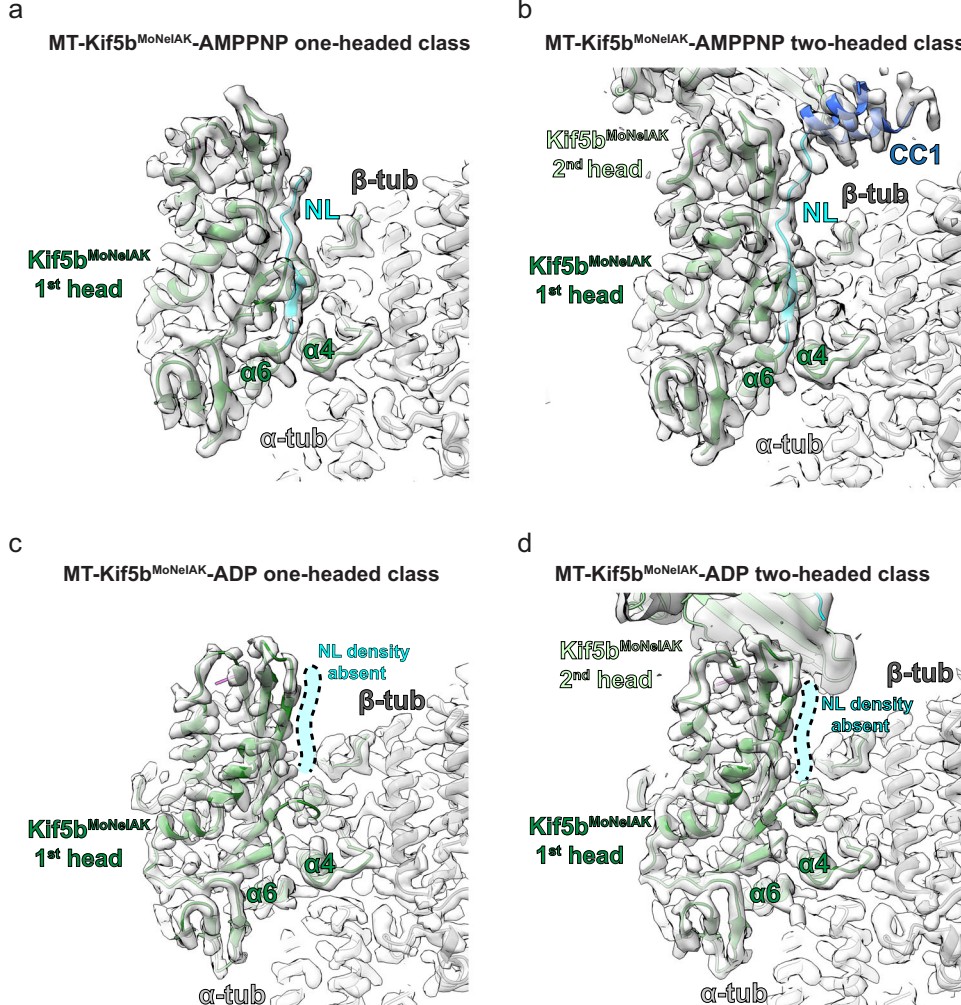

**Fig. 6 | MT-associated Kif5b[MoNeIAK] undocks its neck linker in the presence of ADP. a–d** Views of the neck linker and coiled-coil of the MT-associated head of Kif5b[MoNeIAK] bound to **a, b** AMPPNP or **c, d** ADP in (**a, c**) one- or (**b, d**) two-headed states, with corresponding density in semi-transparent gray. In (**c**) and (**d**), the position where density for a docked neck linker could be expected (but is absent

here) is indicated in semi-transparent cyan. All cryo-EM densities shown in (**a–c**) were filtered using DeepEMhancer[62]. In (**d**), the displayed density for the 1st head was filtered using DeepEMhancer[62], while the density for the 2nd head displayed density was low-pass filtered to 6 Å resolution for resolution-appropriate visualization.

in Lysogeny Broth (LB) supplemented with kanamycin. Small-scaled overnight cultures were used to inoculate large-scale cultures (1:500 ratio) in Terrific Broth (TB) supplemented with 2 mM MgCl$_2$, 0.1% glucose, and antibiotics. Cell cultures were incubated at 37 °C in a shaking rotator until OD600 reached 0.9–1.0. Protein expression was induced with 0.4 mM isopropyl β-D-1-thiogalactopyranoside (IPTG) at

18 °C for 16 h. Cells were harvested by centrifugation ($5000 \times g \times 20$ min, 4 °C) and the cell pellet resuspended in lysis buffer composed of 20 mM Tris pH 8.8, 200 mM NaCl, 2 mM MgCl$_2$, 1 mM EGTA, 1 mM MgATP and 5 mM β-mercaptoethanol (β-ME) supplemented with protease inhibitor cocktail (Merck) and benzonase endonuclease (10 U/ml) (Merck). After homogenization, cells were lysed by

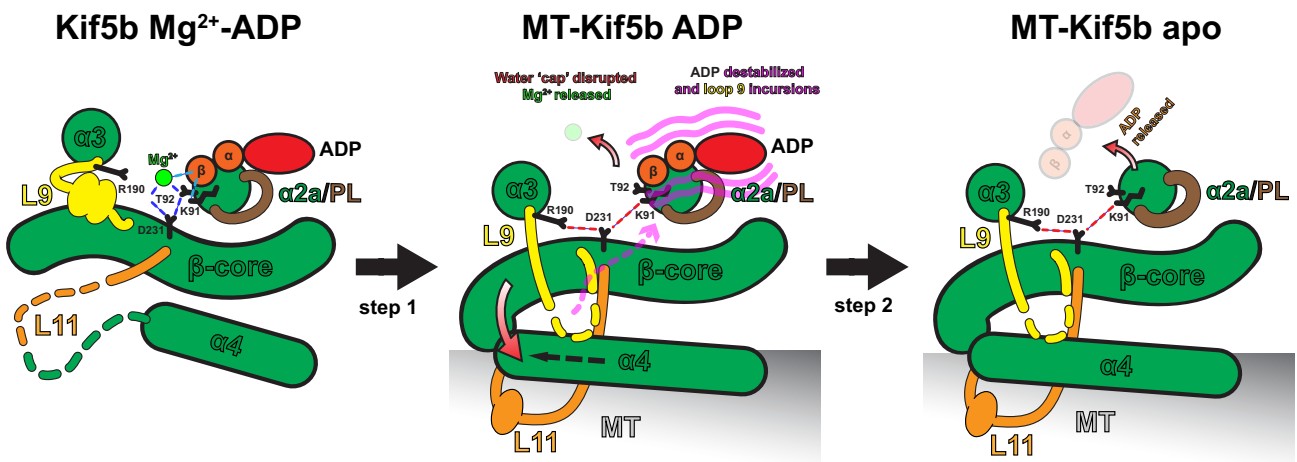

**Fig. 7 | Model for a two-step structural mechanism of MT-stimulated ADP release.** Schematic illustrating a proposed two-step structural mechanism of MT-stimulated ADP release, showing movements of key SSEs and side chains relative to the static helix α2a and P-loop. MT binding triggers transition to an apo-like state, including helix α4 extension and twisting of core β-sheet (β-core) along with movements of overlying helix α3 and loop 9 elements. These movements dismantle a hydrogen bonding network involving D231 and T92 that coordinates $Mg^{2+}$ through a water 'cap' (blue dashed lines). This drives $Mg^{2+}$ out of the complex, with D231 forming a new bonding network with R190 and K91 (red dashed lines). These changes disrupt key bonds stabilizing ADP's β-phosphate (cyan dashed lines), leaving ADP only weakly coordinated by remaining contacts with helix α2a and the P-loop, promoting its subsequent release mediated by loop 9 dynamics. α and β are α-phosphate and β-phosphate, respectively.

sonication on ice, followed by centrifugation ($20,000 \times g \times 45$ min, 4 °C). The soluble material was subsequently filtered using a 0.22 mm pore size filter and was loaded onto a 5 ml HisTrap HP column (GE Healthcare), which was pre-equilibrated with lysis buffer, and immobilized metal affinity chromatography (IMAC) was performed at 4 °C. Target proteins eluted in an imidazole linear gradient were buffer exchanged in lysis buffer, using a dialysis membrane (with 3.5 MWCO) for 3 h. Kif5b$^{MoNeIAK}$ was further incubated overnight at RT with thrombin-conjugated beads (Thrombin CleanCleave Kit, Sigma–Aldrich) for His-tag cleavage. The protein-bead mixture was then filtered using a gravitational column, and the untagged protein was collected by reverse IMAC on a HisTrap HP column (GE Healthcare) pre-equilibrated with lysis buffer. Untagged Kif5b$^{MoNeIAK}$ was buffer exchanged using a PD-10 desalting column in 50 mM Hepes pH 7.5, 20 mM NaCl, 2 mM MgCl$_2$, 1 mM EGTA, 1 mM MgATP, and 5 mM β-ME. The protein was further purified by Ion Exchange Chromatography (IEX) using a RESOURCE S cation exchange column (Cytiva), and the protein was eluted in a salt gradient (50 mM Hepes pH 7.5, 1 M NaCl, 2 mM MgCl$_2$, 1 mM EGTA, 1 mM MgATP, 5 mM β-ME), followed by size-exclusion chromatography (SEC) on a HiLoad 16/600 Superdex 200 column (GE Healthcare) pre-equilibrated with 25 mM HEPES pH 7.5, 150 mM NaCl, 2 mM MgCl$_2$, 1 mM EGTA, 20 μM MgATP, and 5 mM β-ME. Kif5b$^{MoNeXXX}$ was purified by SEC immediately after the IMAC step without His$_6$-tag cleavage using the same buffer conditions employed for Kif5b$^{MoNeIAK}$.

### ATPase assay
Kinetic measurements were carried out with the Enzyme-Linked Inorganic Phosphate ATPase Assay (the ELIPA™ Assay) kit from Cytoskeleton Inc. using the reagents supplied. Motor proteins used in the assay were N-terminally His$_6$-tagged and purified as described above.

### MT preparation for cryo-EM
Purified and lyophilized porcine brain tubulin (>99% pure, Cytoskeleton Inc.) was reconstituted on ice to 10 mg/ml in BRB80 (80 mM PIPES, 2 mM MgCl$_2$, 1 mM EGTA, 1 mM DTT, pH 6.8), then snap frozen in liquid nitrogen and stored at −80 °C. Upon thawing, the tubulin was centrifuged at $11,900 \times g$ on a desktop centrifuge at 4 °C to remove large aggregates, then the supernatant extracted for further use. To polymerize MTs, the supernatant was then diluted to 5 mg/ml tubulin in BRB80 with 1 mM GTP (Sigma) and incubated at 37 °C for 45 min. A solution of 1 mM paclitaxel (Calbiochem) in DMSO was then added and MTs incubated at 37 °C for another 45 min. Stabilized MTs were left at room temperature for at least 24 h before use.

### MT co-sedimentation assays
In $7 \times 20$ mm Thickwall Polycarbonate tubes (Beckman Coulter), paclitaxel-stabilized MTs were mixed at room temperature with Kif5b proteins in BRB80 buffer plus indicated the nucleotides or metal salts. Controls with no MTs or no Kif5b were included. For conditions without $Mg^{2+}$, we added 10 mM EDTA to remove any possible free $Mg^{2+}$ ions. After incubating for 10 min, the samples were centrifuged at $\sim 110,000 \times g$ in an Airfuge® Ultracentrifuge (Beckman Coulter). The supernatants were extracted and mixed 1:1 with 2× SDS-PAGE sample buffer (ThermoFisher Scientific). The pellets were then resuspended with ice-cold BRB80, then further mixed and resuspended with 2× SDS-PAGE sample buffer and extracted. Pellet and supernatant samples were analyzed by SDS-PAGE and stained with InstantBlue® Protein Stain (Expedeon). For densitometric quantification, background-subtracted band intensities were measured from lanes in the Coomassie-stained SDS-PAGE gel with the 'Gel Analyzer' tool in FIJI software[58].

### Sample preparation for cryo-EM
MTs were diluted in BRB80 at room temperature to 0.16 mg/ml before use. Kif5b was diluted in BRB80 to 1 mg/ml and preincubated with 5 mM AMPPNP or ADP for 20 min at 4 °C before use. A volume of 3 μl of MTs was preincubated on glow-discharged C-flat™ carbon-coated copper EM grids (2 μm holes with 2 μm hole spacing, Protochips) at room temperature for 45 s, excess buffer manually blotted away, then 3 μl of Kif5b warmed to room temperature was added for 30 s. Excess buffer was again manually blotted away, followed by another application of 3 μl Kif5b. Grids were then placed in an EM GP automated vitrification device (Leica Microsystems) operating at room

temperature and 80% humidity, incubated for a further 45 s, then blotted and vitrified in liquid ethane.

## Cryo-EM data collection

Cryo-EM data was acquired at the London Consortium for Cryo-EM (LonCEM) on a Krios G3i operating at 300 kV, with a Gatan K3 direct electron detector and a Bioquantum Imaging Filter. Low-dose movies were collected automatically with EPU (v3.4, ThermoFisher) software with a super-resolution mode sampling of 0.54 Å/pixel, in zero-loss imaging mode with a 20 eV energy-selecting slit. A defocus range of 0.7–2.5 μm was used, and the total movie dose was ~52 e⁻/Å² spread over 36 frames, with electron (e⁻) counting at 16 e⁻/physical pixel/second.

## Cryo-EM data processing

Near-identical processing pipelines were used for datasets of ADP-bound Kif5b$^{MoNeXXX}$ or ADP- or AMPPNP-bound Kif5b$^{MoNeIAK}$ (Supplementary Fig. 3). Movie frames were binned ×2 during alignment using Relion v4's implementation of Motioncorr2[59], to generate unweighted and dose-weighted sums at the physical pixel size of 1.08 Å/pixel. Full dose sums were used for CTF determination in CTFFIND v4.1.14[60], then dose-weighted sums were used in particle picking, processing, and generation of the final reconstructions.

MTs were boxed manually in Relion v3.0's helical mode and then 4× binned segments extracted with a box separation distance of 82 Å (roughly the MT dimer repeat distance). From this stage, MT segments were processed with a modified and extended version of MiRP[45] implemented within Relion v3.0. Briefly, a supervised 3D classification was run to undecorated 11–16 protofilament MT references, with the modal class designated within each MT assigned to all segments within that MT. The dominant 13-protofilament class was then selected and aligned to a 12 Å low-pass filtered reference of a 13-protofilament MT decorated with ADP-AlF₄-bound kinesin-1 motor domains, generated from PDBs codes 5syf and 4hna using ChimeraX's (v1.6)[61] 'molmap' command. Rough $x–y$ translations and ψ angles found were then smoothed between adjacent segments along each MT using custom scripts. Rough φ angles were then determined for segments with a second 3D refinement, and the median φ angles for each MT were assigned to all segments in a given MT. Following another round of smoothing $xy$ translations and Euler angle assignments, a fine C1 3D local refinement was then performed with 4x binned segments. 'Segment averages' were then generated by averaging each segment with its 7 adjacent partners within each MT. Using segment averages, assigned φ angles for each MT were checked by supervised 3D classification without alignment to low-pass filtered kinesin-only references rotated and translated to represent all possible seam positions and αβ-tubulin registers (i.e., 26 references for a 13-protofilament MT, with 13 seam positions and their counterparts translated one monomer along the helical axis). Rough final φ angles were assigned according to the modal 3D class of all segments within each MT. Non-binned segments were then reextracted, and a fine C1 3D local refinement was performed, followed by per-particle CTF refinement and Bayesian polishing, each followed by another fine C1 3D local refinement. MT segments were then subjected to an unsupervised 3D classification without alignment using 4 classes, and the highest quality classes selected and inputted into another round of fine C1 3D local refinement.

Symmetry expansion using 13pf MT symmetry parameters determined in Relion was performed. The symmetry-expanded coordinates were then used to extract 1× binned particles (240 pixels at 1.08 Å/pix) focused on the Kif5b$^{MoNeIAK}$ kinesin-tubulin interface of each asymmetric unit. To carry both CTF refinement and polishing information from whole MT segments to individual asymmetric unit sub-particles, a signal subtraction job, but without signal subtraction applied, was used for this extraction step. Using a reference generated

from this extracted asymmetric unit data, a 3D local refinement with a wide circular mask was then performed on these particles. Following this step, a supervised 3D classification without alignment was performed, using references with or without an MT-associated kinesin head. The kinesin-decorated class was selected and then either processed further (for Kif5b$^{MoNeXXX}$ data and Kif5b$^{MoNeIAK}$ data when combining one and two-headed states for maximum resolution of the MT-bound head) or reextracted focused on the tail density in each asymmetric unit (for separate processing of Kif5b$^{MoNeIAK}$ one and two-headed states focused on the tail region). Particles focused on the kinesin-tubulin interface were subjected to a final local 3D refinement using a soft mask inclusive of the tubulin dimer and MT-associated motor domain. Kif5b$^{MoNeIAK}$ particles extracted and focused on the tail were subjected to an unsupervised 3D classification without alignment using a soft mask inclusive of the distal motor head. Two classes, one with and one without the second motor head, were found. The one-headed class was then subjected to a final local 3D refinement using a soft mask inclusive of the tubulin dimer and MT-associated motor domain. The two-headed class was then subjected to a final local 3D refinement using a soft mask inclusive of either tubulin and both motor domains or just both motor domains without tubulin.

Global resolutions were estimated at the gold-standard FSC 0.143 cut-off (noise-substitution test-corrected), using soft masks around the refined density. Local resolutions were estimated using Relion's inbuilt software. Reconstructions used for display were locally sharpened using DeepEMhancer[62], Relion's local resolution filtering, or LocScale[63]. Final particle numbers are shown in (Supplementary Table 1).

## Pseudo-atomic model building

Initial homology models of AMPPNP or ADP-bound human Kif5b$^{MoNeIAK}$ and Kif5b$^{MoNeXXX}$ MT-associated heads were created using MODELLER (web service)[64] and template X-ray structures of a kinesin-1 motor domain bound to ADP-AlF₄ (PDB code 4hna) or ADP (PDB code 4lnu), respectively. A taxol-bound tubulin dimer (PDB code 5syf) was added to these models. To make the Kif5b$^{MoNeIAK}$ two-headed state with ADP, a 2nd head was created using MODELLER and a combination of ADP-bound kinesin-1 structures (PDB codes 1bg2 and 2y65). To make the Kif5b$^{MoNeIAK}$ two-headed state with AMPPNP, a 2nd head was created using MODELLER and a combination of AMPPNP-bound kinesin-3 (PDB code 1vfv) and ADP-bound kinesin-1 (PDB code 2y65) motor domain crystal structures. The tail was modeled based on the *Drosophila melanogaster* kinesin-1 autoinhibited dimer crystal structure (PDB code 2y65). After trimming the models to remove disordered regions not represented by EM density, these models were rigid-fitted into cryo-EM density with the 'fit-in-map' ChimeraX tool. Iterative rounds of Coot (v0.9.8.92)[65] and real space refinement in Phenix (v1.20.1_4487)[66] were then performed, with low-resolution settings applied to the 2nd head to ensure its conservative modeling given its lower resolution density. For Kif5b$^{MoNeIAK}$ single-headed states, the 2nd head was removed and part of the tail was deleted, followed by a further round of real-space refinement. Cryo-EM density and model analysis, and display used ChimeraX. Model refinement statistics are presented in (Supplementary Table 1). All model refinement was performed with density maps sharpened according to local resolution in the Relion software.

## Molecular Dynamics simulations

The crystallographic ADP-bound/Mg²⁺-loaded Kif5b motor structure solved in the absence of MTs (PDB code 1bg2) was used to model the MT-free system. The MT-Kif5b$^{MoNeXXX}$-ADP structure presented here was used for the MT-bound/ADP-bound/Mg²⁺-free system. Missing residues were modeled in ChimeraX (1.7.1) using the MODELLER (10.5) add-on. The CHARMM-GUI[67] (web service) was used to generate the topology of both systems using the CHARMM36m force field[68], followed by solvation and ionization of the system (Na⁺ and Cl⁻ ions) in a

cubic simulation box. This resulted in a system of 97,319 and 324,476 atoms for the MT-free and MT-bound systems, respectively. The latter system is larger as it includes the tubulin dimer. The ionic strength was set to 0.15 M, and the temperature was set to 300 K. The simulations were run using GROMACS (2022.3)[69], with md integrator, the PME scheme, and the NVT setup. To preserve the starting orientation of the tubulin units, a set of positional restraints on their backbone atoms and cofactors was applied, with a harmonic force constant of 400 kJ mol$^{-1}$ nm$^{-1}$.

Three independent replicas of 1 μs were initially run for both constructs. We refer to this as the Level 1 dataset. For the MT-bound system, a new dataset was generated (Level 2) by selecting the two frames with the highest combined value of RMSD and deviation from the starting pair distances E199-S98, T87-ADP, S88-ADP, S89-ADP and K91-ADP (see Supplementary Fig. 14). For each of these, five replicas of 150 ns were run. A further Level 3 dataset was generated starting from seven frames, selected using the same criteria adopted previously. Five replicas of 150 ns for each frame were then computed. RMSD (after Cα alignment of residues 75–95 of the motor domain) and pair distances along all MD trajectories were calculated with VMD (1.9.3)[70].

### Reporting summary
Further information on research design is available in the Nature Portfolio Reporting Summary linked to this article.

## Data availability
MT-bound Kif5b$^{MoNeXXX}$ and Kif5b$^{MoNeIAK}$ one and two-headed asymmetric unit reconstructions are deposited within the Electron Microscopy Data Bank (EMDB) alongside corresponding atomic models deposited in the Protein Data Bank (PDB) as follows; MT-Kif5b$^{MoNeIAK}$-AMPPNP one-headed class: EMD-19174, PDB 8RHB, MT-Kif5b$^{MoNeIAK}$-AMPPNP two-headed class: EMD-19176, PDB 8RHH, MT-Kif5b$^{MoNeIAK}$-ADP one-headed class: EMD-19188, PDB 8RIK, MT-Kif5b$^{MoNeIAK}$-ADP two-headed class: EMD-19192, PDB 8RIZ, MT-Kif5b$^{MoNeXXX}$-ADP: EMD-51477, PDB 9GNQ. EMDB and PDB accession codes are also shared in Supplementary Table 1. Topology and input files of the molecular dynamics trajectories are available on Zenodo under https://doi.org/10.5281/zenodo.15050784. Source data are provided with this paper.

## Code availability
The extended version of the MiRP package used during the processing of cryo-EM image data can be found at https://github.com/AthertonLabKCL/

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

## Acknowledgements

J.A. and T.F. are supported by the Biotechnology and Biological Sciences Research Council (BBSRC) grant BB/V006568/1. M.S.C. was supported by the BBSRC grant BB/S000828/1 awarded to R.A.S. E.P. and M.L. are supported by the Italian Ministry for Universities and Research (MUR) PRIN-PNRR grant P2022LSH5A, awarded to G.M. and R.A.S. L.S.P. is supported by the Italian Ministry for Universities and Research (MUR) PRIN grant 2022ERB7SL, awarded to R.A.S. M.D.P. is a PhD student funded by the Italian Ministero dell'Università e della Ricerca (MUR). We acknowledge the contribution of two Master's students in the RAS group at KCL, Miss Miral Tariq and Mr Henry Cornish, in the early phase of this work. The London Consortium for Electron Microscopy (LonCEM) is supported by the Wellcome Trust grant 206175/Z/17/Z and its partner institutes. We acknowledge ISCRA (IsB29_ESMP) for awarding this project access to the LEONARDO supercomputer, owned by the EuroHPC Joint Undertaking, hosted by CINECA (Italy). We also acknowledge access and services provided by the National Facility for Structural Biology – IU3, Fondazione Human Technopole, Milan, Italy; Call for Access 24-SB-PILOT, Project ID1777959.

## Author contributions

J.A., M.S.C., M.D.P., T.F., E.P., S.P., and L.S.P. carried out the experiments. M.L., M.V.A.M., and G.M. performed molecular dynamics simulations. J.A. and R.A.S. conceived the study, designed the experiments, interpreted the results, and wrote the paper with contributions from all authors.

## Competing interests

The authors declare no competing interests.
