## [Transparent Peer Review file · Nature Communications]

Microtubule association induces a Mg-free apo-like ADP pre-release conformation in kinesin-1 that is unaffected by its autoinhibitory tail

Corresponding Author: Professor Roberto Steiner

Version 0:

Reviewer comments:

Reviewer #1

(Remarks to the Author)

Kinesin-1/KIF5 is a well-studied MT-based motor protein. In this manuscript, Atherton et al used cryo-EM techniques to solve the high-resolution structure of the MT-bound KIF5b motor head. They found that the motor domain is bound to ADP while closely resembling the nucleotide-free conformation, and the nucleotide-binding site lacks a Mg²⁺ ion, which typically coordinates with ADP. They then proposed that this Mg²⁺-free structure presents an intermediate state during the MT-stimulated ADP release, an important step in the motor's mechanochemical cycle. The authors also determined KIF5b motor-MT structures with the bound IAK motif. The manuscript provides many detailed structural insights by carefully comparing KIF5b-MT structures in different states with those reported in the literature. However, previous similar observations in kinesin structures, especially KIF14-MT structures, somehow temper the novelty of this study. Also, some findings or conclusions in the present manuscript lack experimental verification.

Detailed comments:

- 1) The main discovery of this manuscript is the mechanism of MT-stimulated ADP release in KIF5b, as shown in Fig.7. However, a similar Mg²⁺-free structure of the motor-ADP-MT complex has already been reported in an extensive Cryo-EM characterization of KIF14 (Benoit et al, Nat Commun 2021). The overall changes induced by MT binding, including those sidechain orientations, are highly similar to those observed in KIF14 (Fig.S11). Given these similarities, what additional key insights can the Mg²⁺-free KIF5b-MT structure provide that advance our understanding learned from KIF14?
- 2) It appears that the motor-MT complex can be prepared with the motor head alone without the autoinhibitory IAK tail. Why did the authors choose to use a chimeric construct with a fused mutated IAK sequence? Using the motor head directly could have simplified the structural analysis. It is important to consider that this mutated sequence, even if no longer binding to the motor, might still have nonspecific effects on the motor's activity. To address this, at least, the ATPase activity of these fusion proteins should be measured and compared to that of the motor head alone, showing whether this artificial fusion matters.
- 3) Another related concern. The authors claim that ADP will dissociate from the motor head in the absence of Mg²⁺ coordination. However, the current evidence is not rigorous enough to confirm their hypothesis. The only experiment (Fig.3c) was performed using the fusion protein, which as mentioned above may have nonspecific effects. More convincing evidence should be provided. One possibility is to conduct molecular simulations to capture the whole process of KIF5 motor binding to MTs. Such simulations would not only elucidate the MT binding-induced Mg²⁺ release followed by ADP dissociation but also possibly explain the molecular determinants driving these dynamic changes, which are currently lacking in the manuscript.
- 4) The authors observed that the IAK-fused motor head can bind to MTs in both one- and two-headed states, with a higher abundance of the one-head state in the AMPPNP condition compared to the ADP condition (Fig. 1i). Does the AMPPNP state has a higher tendency to form IAK-mediated dimer? Or this is an artifact of data processing? The authors could test the influence of different nucleotide states on the two-headed state formation. For example, compare the dimer fraction of the motor domain with the addition of the IAK peptide using analytical SEC under different nucleotide conditions. Also, the description of the cryo-EM workflow (Fig.S2) is not sufficiently detailed. More classification information, like picked particle number and percentage of each class, should be presented for reader evaluation.

5) The proposed model in Fig.S15 lacks evidence support. The presence of two tails is more likely to enhance avidity and thereby ensure effective autoinhibition. The authors should either provide evidence or remove it from the manuscript.

6) How does the head release the IAK motif following KIF5 motor activation when the MT-associated head retains the capacity to bind this autoinhibitory sequence?

(Remarks on code availability)

Reviewer #2

(Remarks to the Author)

This paper by Atherton et al. is focused on determining the role of IAK motif in the kinesin-1 mechanochemical cycle. The work directly addresses the confusing and conflicting results regarding the role of the kinesin-1 tail in kinesin inhibition and stepping. To address this, the authors created a 'bonsai' version of kinesin-1 which included the motor domain, neck linker, neck coil, and the IAK motif. Additionally, they show a structure of an ADP-bound motor domain without a tail to indicate structural changes caused by the IAK motif.

The structures determined are high quality and provide new insights into the biphasic model of ADP release and IAK motif binding to motors on microtubules. The interpretations of the structures and comparison to published work is well thought out and clearly shown to the reader. The figures nicely provide the reader with both the model and maps to aid interpretation. The only comments are minor related to data interpretation or text descriptions.

Minor comments:

-Modeling low resolution areas: In the two headed states, the resolution is lower for the motor held away from the microtubule. While the overall docking of the kinesin is OK, the assignment of nucleotide state to this motor appears to be overinterpreted. The density does not appear to be unambiguously able to identify the nucleotide state. Considering that this is not a major point of the paper, I would suggest the authors do not model the nucleotide unless they have data on nucleotide side for the second motor head.

-“high-resolution cryo-EM structures” should be “cryo-EM structures.” High resolution is subjective and could be 1 Å, 2 Å, 4 Å, etc. Please remove this from the manuscript. You can make relative claims, such as these are higher in resolution than previous maps.

-“Fig.11” on page 6 - there is no main Figure 11 in the text.

-The deposited model 8RHB does not fit exactly within the associated map EMD-19174. It appears to be slightly shifted. Could the authors address this discrepancy in the deposition?

-“R203 of switch I (in loop 9) and E236 of switch II (in loop 11)” - the density for E236 is not as strong as the R203 density, suggesting flexibility of E236. Can the authors remove this indication of a hydrogen bond between these two side chains since it is not supported by the density map? The remainder of the paragraph describing other differences in kinesin structures is OK. This is related to a paragraph on page 7 that references Figure 2e.

- Fig 1n shows poor density for the tail for the MoNeIAK-ADP one-head state. To show the reader that the density is real, could you show a comparison in the same figure to the MoNeXXX-ADP one-headed state from Fig 2a?. All other panels have unambiguous density - beautiful!

-Could you clarify in figure legends which figures and maps are DeepEMhancer, LocSpiral, or RELION local resolution filter? These different filtering tools are mentioned in the methods but not clearly indicated throughout the manuscript.

-Were the models refined Deepemhancer maps? If so, please redo refinement using tools that are not generative artificial intelligence filters. This point was unclear in the methods.

(Remarks on code availability)

N/A

Version 1:

Reviewer comments:

Reviewer #1

(Remarks to the Author)

The authors have done an excellent job in addressing my concerns. I support their publication in Nature Communications.

(Remarks on code availability)

Reviewer #2

(Remarks to the Author)

Thank you for addressing our comments.

(Remarks on code availability)

n/a

We thank both reviewers for the time spent reviewing our manuscript. We are pleased with the very supportive comments from Reviewer #2. We also thank Reviewer #1 for giving us the opportunity to clarify some of the key findings presented in this work.

Reviewer #1 (Remarks to the Author):

Kinesin-1/KIF5 is a well-studied MT-based motor protein. In this manuscript, Atherton et al used cryo-EM techniques to solve the high-resolution structure of the MT-bound KIF5b motor head. They found that the motor domain is bound to ADP while closely resembling the nucleotide-free conformation, and the nucleotide-binding site lacks a Mg²⁺ ion, which typically coordinates with ADP. They then proposed that this Mg²⁺-free structure presents an intermediate state during the MT-stimulated ADP release, an important step in the motor's mechanochemical cycle. The authors also determined KIF5b motor-MT structures with the bound IAK motif. The manuscript provides many detailed structural insights by carefully comparing KIF5b-MT structures in different states with those reported in the literature. However, previous similar observations in kinesin structures, especially KIF14-MT structures, somehow temper the novelty of this study. Also, some findings or conclusions in the present manuscript lack experimental verification.

In this opening comment the reviewer suggests that our work is tempered in its novelty because of some structural similarities with Kif14-MT structures. We disagree with this general statement. We explain why in response to the reviewer's detailed comment 1) that mentions specifically Kif14.

Detailed comments:

1) The main discovery of this manuscript is the mechanism of MT-stimulated ADP release in KIF5b, as shown in Fig.7. However, a similar Mg²⁺-free structure of the motor-ADP-MT complex has already been reported in an extensive Cryo-EM characterization of KIF14 (Benoit et al, Nat Commun 2021). The overall changes induced by MT binding, including those sidechain orientations, are highly similar to those observed in KIF14 (Fig.S11). Given these similarities, what additional key insights can the Mg²⁺-free KIF5b-MT structure provide that advance our understanding learned from KIF14?

The work of Benoit et al. (2021) deals with Kif14, an unconventional mitotic kinesin with several atypical structural and functional features. In contrast, here we studied Kif5 (also known as conventional kinesin), the prototypical kinesin motor that enables processive cargo transport in interphase cells.

Some of the properties that distinguish Kif14 from other kinesin members include: i) an unusual ADP-bound motor domain structure in the absence of MTs, where Mg²⁺ is absent and an apo-like conformation is adopted (characterized by an extended helix α 4, unfurled loop 9, and a twisted core β -sheet; PDB code 4OZQ); ii) an exceptionally high MT affinity in the presence of ADP; iii) an unusually high basal ATPase rate in the absence of MTs; iv) weak stimulation of ATPase activity by MTs.

None of the above features are shared with Kif5 or, to our knowledge, any other kinesin family members. Given Kif14's unconventional features, it was therefore unclear whether the Kif14 MT- and ADP-bound structure could be considered representative of the family or an isolated case. Notably, the apo-like, Mg²⁺-free ADP-bound conformation of Kif14 implies a fundamentally different ADP-release transition compared to KIF5b. Furthermore, since MTs are not required to stimulate Mg²⁺ release in Kif14, the established two-step ADP release mechanism typical of conventional kinesins and discussed in our work does not

apply to this atypical kinesin. Presumably for this reason, the implications of the Kif14 structure for the two-step ADP release mechanism — including Mg^{2+} release — in conventional kinesins are not addressed in Benoit et al. (2021). Our structural work fills this substantial gap in knowledge by providing a detailed atomistic model for the two-step ADP release process in Kif5 thus significantly advancing our mechanistic understanding of conventional kinesins. We are grateful for this reviewer's suggestion to also perform MD simulations (see answer to 3) below), as the computational work further strengthened our mechanistic interpretation.

We feel we should also point out that the mechanism of MT-stimulated ADP release in conventional kinesin is one of the main discovery points of our work. In addition to this, our study reveals several other key insights:

1. we show unambiguously that in MT-bound kinesin-1, the tail binds at a site distal to the switch loops/nucleotide-binding pocket challenging a previous lower-resolution cryo-EM study, (Dietrich et al., 2008, PNAS) that relied on chemical cross-linking of the tail to the motor leading to the proposal that the tail directly inhibits ADP release;
2. we show unambiguously that the kinesin-1 tail does not prevent neck-linker undocking in MT-bound kinesin, contradicting the previously proposed 'double-lockdown' model of autoinhibition (Kaan et al., Science, 2011);
3. we show unambiguously that the kinesin-1 tail has minimal effect on the conformation of the MT-bound motor domain in ADP or ATP-mimic states, discounting several earlier hypotheses about the tail's role in kinesin autoinhibition (Yonekura et al., Biochemical and Biophysical Research Communications, 2006; Hackney & Stock, Nature Cell Biology, 2000; Seiler et al., 2000; Nature Cell Biology; Coy et al., Nature Cell Biology, 1999; Stock et al., JBC, 1999).

Overall, our findings present a significant degree of novelty and advance our understanding of kinesin-mediated cytoskeletal transport.

2) It appears that the motor-MT complex can be prepared with the motor head alone without the autoinhibitory IAK tail. Why did the authors choose to use a chimeric construct with a fused mutated IAK sequence? Using the motor head directly could have simplified the structural analysis. It is important to consider that this mutated sequence, even if no longer binding to the motor, might still have nonspecific effects on the motor's activity. To address this, at least, the ATPase activity of these fusion proteins should be measured and compared to that of the motor head alone, showing whether this artificial fusion matters.

A key goal of our study was to analyze the specific effects of the autoinhibitory IAK-motif located within the C-terminal tail in the context of MT binding. To mimic the intramolecular interaction relevant to full-length kinesin-1, we performed our cryo-EM analysis using the Kif5b^{MoNeIAK} fusion construct. Thus, we elected to carry out the 'control experiment' (i.e. absence of the IAK-motif) using exactly the same 'background', hence the use of the chimeric Kif5b^{MoNeXXX} construct.

In consideration of the reviewer's request, we have generated a novel construct (Kif5^{MoNe}) that corresponds to the motor alone without the extension. We have carried out ATPase assays using Kif5^{MoNeXXX} as well as Kif5^{MoNe} and the two proteins display identical kinetic parameters within error. This indicates that motor-MT complex is not affected by the disordered extension. These results have now been added as a new supplementary figure with four panels (Supplementary Figure 2 in the revised manuscript). Accordingly, we have also amended the text of the 'Bonsai kinesin-1 chimeras for cryo-EM studies' section (in the

*changes_highlighted.pdf files uploaded, added text is in green and ~~deleted text~~ is in red with strikethrough).

3) Another related concern. The authors claim that ADP will dissociate from the motor head in the absence of Mg²⁺ coordination. However, the current evidence is not rigorous enough to confirm their hypothesis. The only experiment (Fig.3c) was performed using the fusion protein, which as mentioned above may have nonspecific effects. More convincing evidence should be provided. One possibility is to conduct molecular simulations to capture the whole process of KIF5 motor binding to MTs. Such simulations would not only elucidate the MT binding-induced Mg²⁺ release followed by ADP dissociation but also possibly explain the molecular determinants driving these dynamic changes, which are currently lacking in the manuscript.

In our manuscript we reference previous biophysical studies on kinesin-1 that demonstrate that removal of Mg²⁺ strongly stimulates ADP release (Cheng et al., 1998; Hackney & Stock 2008). Thus, our cryo-EM work provides a much-needed structural context for these observations. However, we acknowledged the reviewer's point and decided to perform molecular dynamics (MD) simulations. This was a significant effort, but we are happy that we did it as it substantially strengthens our cryo-EM results and mechanistic proposal. We thank the reviewer for this suggestion.

Our MD work is now discussed in a novel section entitled: "ADP dissociates from the MT-bound/Mg²⁺-free Kif5 motor in Molecular Dynamics simulations" and is accompanied by ten additional panels in Fig.3 and two new supplementary figures (Supplementary Fig. 13 and 14 in the revised manuscript). Briefly, the main structural differences that accompany the transition from the MT-free to MT-bound state are: i) the extension of the MT-interacting α 4 helix alongside the stabilization of loop 11, ii) twist of the core β -sheet and accompanying readjustments in the relative positions of the kinesin subdomains and iii) the loss of the helical portion of loop 9 that becomes partially disordered. MD shows that these have a significant impact on the dynamics of the ADP site as the longer α 4 helix not only sterically destabilizes loop 9 but it also contributes to its relocation in proximity of the catalytic site. Here, loop 9, directly displaces ADP with E199 transiently occupying the phosphate pocket. In addition to the novel figures that highlight the MD work we have also uploaded a video in the supplementary material that shows ADP release from the MT-bound/Mg²⁺-free structure.

As reported in the Data Availability section, topology and input files of the MD trajectories have also been made available on Zenodo under doi: [10.5281/zenodo.15050784](https://doi.org/10.5281/zenodo.15050784)

4) The authors observed that the IAK-fused motor head can bind to MTs in both one- and two-headed states, with a higher abundance of the one-head state in the AMPPNP condition compared to the ADP condition (Fig. 1i). Does the AMPPNP state has a higher tendency to form IAK-mediated dimer? Or this is an artifact of data processing? The authors could test the influence of different nucleotide states on the two-headed state formation. For example, compare the dimer fraction of the motor domain with the addition of the IAK peptide using analytical SEC under different nucleotide conditions.

The reviewer interpreted (or just wrote) this the wrong way round – Figure 1i shows that in our cryo-EM structures there is a higher abundance of the one-head state in the ADP condition compared to the AMPPNP condition. It is not an artifact of data processing: it simply reflects the relative amounts of the dimer/monomer populations on MTs under the conditions used to prepare the cryo-EM samples. Due to the nature of the blotting process, there is limited reproducibility and therefore it is impossible to ascertain the exact final

effective concentration on the grids. As shown in Figure 1d, SEC runs (obviously without MTs) clearly show that Kif5b^{MoNeI_{AK}} can dimerize whilst Kif5b^{MoNe^{XXX}} does not. This appears rather uncontroversial given the IAK motif-dependent cross-linking of the motors observed here by cryo-EM and previously by X-ray crystallography.

Also, the description of the cryo-EM workflow (Fig.S2) is not sufficiently detailed. More classification information, like picked particle number and percentage of each class, should be presented for reader evaluation.

We have produced a new, more detailed, picture highlighting the cryo-EM workflow, including class percentages. This is now presented in Supplementary Fig. 3. We have also updated Table 1 with more information, as requested.

5) The proposed model in Fig.S15 lacks evidence support. The presence of two tails is more likely to enhance avidity and thereby ensure effective autoinhibition. The authors should either provide evidence or remove it from the manuscript.

Supplementary Fig. 15 in the original version of the manuscript did not dispute the fact that the presence of two tails ensures effective autoinhibition. However, as it was not essential, we have removed it.

6) How does the head release the IAK motif following KIF5 motor activation when the MT-associated head retains the capacity to bind this autoinhibitory sequence?

This is an interesting question that is, however, outside the scope of this study. Whilst the structural mechanism of release from autoinhibition remains unclear, there is some proposal that additional trans-acting factors such as cargo adaptors and MAP7 may be involved (Tan et al., 2023).

Reviewer #2 (Remarks to the Author):

This paper by Atherton et al. is focused on determining the role of IAK motif in the kinesin-1 mechanochemical cycle. The work directly addresses the confusing and conflicting results regarding the role of the kinesin-1 tail in kinesin inhibition and stepping. To address this, the authors created a 'bonsai' version of kinesin-1 which included the motor domain, neck linker, neck coil, and the IAK motif. Additionally, they show a structure of an ADP-bound motor domain without a tail to indicate structural changes caused by the IAK motif.

The structures determined are high quality and provide new insights into the biphasic model of ADP release and IAK motif binding to motors on microtubules. The interpretations of the structures and comparison to published work is well thought out and clearly shown to the reader. The figures nicely provide the reader with both the model and maps to aid interpretation. The only comments are minor related to data interpretation or text descriptions.

We thank Reviewer #2 for his/her positive comments on our manuscript.

Minor comments:

-Modeling low resolution areas: In the two headed states, the resolution is lower for the motor held away from the microtubule. While the overall docking of the kinesin is OK, the assignment of nucleotide state to this motor appears to be overinterpreted. The density does not appear to be unambiguously able to identify the nucleotide state. Considering that this is

not a major point of the paper, I would suggest the authors do not model the nucleotide unless they have data on nucleotide side for the second motor head.

While resolution in the 2nd head (motor held away from the MT) is lower, it provides sufficient information to tell whether a nucleotide is present or not. Considering the nucleotide used in sample preparation was either ADP, or the non-hydrolysable ATP analogue AMPPNP, we can place these nucleotides in their respective densities. Modelling of the 2nd head was more conservative than modelling of the 1st head due to its lower resolution-starting models were fitted and conservatively refined in real space with *Phenix.refine* using low resolution settings, preserving the overall secondary and tertiary structures of the starting models. We have added extra text in the 'Cryo-EM data processing' Methods section to clarify this.

-“high-resolution cryo-EM structures” should be “cryo-EM structures.” High resolution is subjective and could be 1Å, 2Å, 4Å, etc. Please remove this from the manuscript. You can make relative claims, such as these are higher in resolution than previous maps.

Agreed. We have modified the text accordingly throughout.

-“Fig.11” on page 6 - there is no main Figure 11 in the text.

On page 6, this is Fig. 1l (lima), not Fig. 11.

-The deposited model 8RHB does not fit exactly within the associated map EMD-19174. It appears to be slightly shifted. Could the authors address this discrepancy in the deposition?

This has been addressed, with a new shifted model uploaded under this PDB code.

-“R203 of switch I (in loop 9) and E236 of switch II (in loop 11)” - the density for E236 is not as strong as the R203 density, suggesting flexibility of E236. Can the authors remove this indication of a hydrogen bond between these two side chains since it is not supported by the density map? The remainder of the paragraph describing other differences in kinesin structures is OK. This is related to a paragraph on page 7 that references Figure 2e.

In cryo-EM, density for acidic side chains is often poor regardless of structural flexibility, due to their increased sensitivity to electron damage and/or an effect of the negative scattering factor of charged oxygen. However, given that there is uncertainty as to why density for E236 is weaker, the hydrogen bond in question has been removed as suggested and the text on page 7 that discusses this interaction modified accordingly.

- Fig 1n shows poor density for the tail for the MoNeIAK-ADP one-head state. To show the reader that the density is real, could you show a comparison in the same figure to the MoNeXXX-ADP one-headed state from Fig 2a?. All other panels have unambiguous density - beautiful!

The most direct comparison of density between MoNeIAK-ADP and MoNeXXX-ADP is shown already, when comparing Figure 1 panels h and j, which display unfiltered density and demonstrate the lack of density for the tail in the later. However, we have added an inset in panel n showing a lack of density for the tail in the MoNeXXX-ADP reconstruction at the same threshold (with the same filtering method) as the density shown for the MoNeIAK-ADP reconstruction.

-Could you clarify in figure legends which figures and maps are DeepEMhancer, LocSpiral, or RELION local resolution filter? These different filtering tools are mentioned in the methods but not clearly indicated throughout the manuscript.

As requested, the filtering/sharpening method used in each figure panel has been clarified in the figure legends.

-Were the models refined Deepemhancer maps? If so, please redo refinement using tools that are not generative artificial intelligence filters. This point was unclear in the methods.

The models were refined using local resolution sharpened maps and not using AI filters. This has been clarified in the Methods text.

We believe we have addressed all relevant points raised by the reviewers and that you will accept our revised manuscript for publication in Nature Communications.

(Note: in the revised version we have additional authors compared to the original submission. This is because of the supplementary experiments and calculations carried out in response to Reviewer 1's comments.)

References

Benoit, M. *et al.* Structural basis of mechano-chemical coupling by the mitotic kinesin KIF14. *Nat Commun* **12**, 3637 (2021).

Cheng, J. Q., Jiang, W. & Hackney, D. D. Interaction of mant-adenosine nucleotides and magnesium with kinesin. *Biochemistry* **37**, 5288-5295 (1998).

Coy, D. L., Hancock, W. O., Wagenbach, M. & Howard, J. Kinesin's tail domain is an inhibitory regulator of the motor domain. *Nat Cell Biol* **1**, 288-292 (1999).

Dietrich, K. A. *et al.* The kinesin-1 motor protein is regulated by a direct interaction of its head and tail. *Proc Natl Acad Sci U S A* **105**, 8938-8943 (2008).

Hackney, D. D. & Stock, M. F. Kinesin's IAK tail domain inhibits initial microtubule-stimulated ADP release. *Nat Cell Biol* **2**, 257-260 (2000).

Kaan, H. Y., Hackney, D. D. & Kozielski, F. The structure of the kinesin-1 motor-tail complex reveals the mechanism of autoinhibition. *Science* **333**, 883-885 (2011).

Seiler, S. *et al.* Cargo binding and regulatory sites in the tail of fungal conventional kinesin. *Nat Cell Biol* **2**, 333-338 (2000).

Stock, M. F. *et al.* Formation of the compact conformer of kinesin requires a COOH-terminal heavy chain domain and inhibits microtubule-stimulated ATPase activity. *J Biol Chem* **274**, 14617-14623 (1999).

Tan, Z. *et al.* Autoinhibited kinesin-1 adopts a hierarchical folding pattern. *Elife* **12** (2023).

Yonekura, H. *et al.* Mechanism of tail-mediated inhibition of kinesin activities studied using synthetic peptides. *Biochem Biophys Res Commun* **343**, 420-427 (2006).